# Empowering LLM Agents with Zero-Shot Optimal Decision-Making through Q-learning

**Jiajun Chai** [1,2], **Sicheng Li** [1,2], **Yuqian Fu** [1,2], **Dongbin Zhao** [1,2,*], **Yuanheng Zhu** [1,2,*]

[1] Institution of Automation, Chinese Academy of Sciences
[2] School of Artificial Intelligence, University of Chinese Academy of Sciences
{chaijiajun2020, lisicheng2024, fuyuqian2022, yuanheng.zhu,
dongbin.zhao}@ia.ac.cn

## Abstract

Large language models (LLMs) are trained on extensive text data to gain general comprehension capability. Current LLM agents leverage this ability to make zero- or few-shot decisions without reinforcement learning (RL) but fail in making optimal decisions, as LLMs inherently perform next-token prediction rather than maximizing rewards. In contrast, agents trained via RL could make optimal decisions but require extensive environmental interaction. In this work, we develop an algorithm that combines the zero-shot capabilities of LLMs with the optimal decision-making of RL, referred to as the **M**odel-based **L**LM **A**gent with **Q**-Learning (MLAQ). MLAQ employs Q-learning to derive optimal policies from transitions within memory. However, unlike RL agents that collect data from environmental interactions, MLAQ constructs an imagination space fully based on LLM to perform imaginary interactions for deriving zero-shot policies. Our proposed UCB variant generates high-quality imaginary data through interactions with the LLM-based world model, balancing exploration and exploitation while ensuring a sub-linear regret bound. Additionally, MLAQ incorporates a mixed-examination mechanism to filter out incorrect data. We evaluate MLAQ in benchmarks that present significant challenges for existing LLM agents. Results show that MLAQ achieves a optimal rate of over 90% in tasks where other methods struggle to succeed. Additional experiments are conducted to reach the conclusion that introducing model-based RL into LLM agents shows significant potential to improve optimal decision-making ability. Our interactive website is available at this link.

## 1 Introduction

Large language models (LLMs) exhibit impressive comprehension capabilities and are widely utilized to address decision-making tasks as an LLM agent(Guo et al., 2024) in a **zero- or few-shot manner**, meaning that the agents make decisions without or with minimal interactions with the environment. In this work, we aim to further empower LLM agents with optimal decision-making via reinforcement learning (RL). Most existing LLM agents improve decision-making through prompt optimization (Wei et al., 2022; Mandi et al., 2023), which heavily relies on the basic LLMs, lacking the ability to maximize rewards. In contrast, recent studies have introduced the Markov Decision Process (MDP) framework to leverage MDP-based planning (Hao et al., 2023). Figure 1 (a) illustrates a typical framework for MDP-based LLM agents, in which the domain description is a manual in natural language to explain the problem to be solved (Yao et al., 2023a). For clarifications, a **domain** represents an environment characterized by unique state and action spaces (e.g., block cubes and robotic arm control are different domains), while a **task** within a domain entails an original-target state pair (e.g., assembling a beef sandwich and a bacon sandwich are different tasks). Among existing LLM agents, the most common framework combines an Monte Carlo Tree Search (MCTS (Kocsis & Szepesvári, 2006))-**Planner** with a **Replay Buffer** to derive the optimal policy (Zhang et al., 2024d). The replay buffer is task-specific, which stores transitions within the same task, sourced either from the environmental interaction (Murthy et al., 2023; Ding et al., 2023) or the imaginary interaction

---

*Corresponding Authors

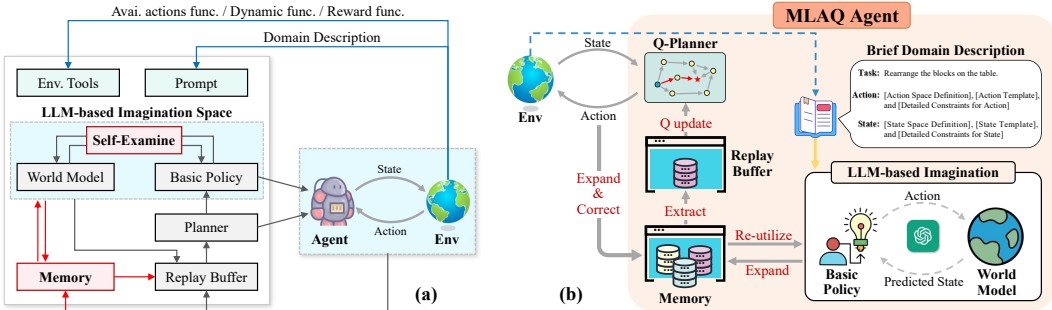

Figure 1: (a) A typical framework of MDP-based LLM agent, where the gray arrows represent processes used by most MDP-based LLM agents and the red arrows and blocks depict processes and modules unique to MLAQ. (b) The overall MLAQ framework. MLAQ interacts with the environment through the Q-Planner, which is supported by the domain-specific memory that extracts a task-specific replay buffer for Q-Update.

(Hao et al., 2023; Yao et al., 2023a) with world model. The concept of the imagination space is derived from Dreamer Hafner et al. (2020), consisting of an **Basic Policy** to generate actions and an **World Model** to predict next states. In this work, both of them is based on LLM, and their interactions could generate imaginary transitions, reducing the demand for environmental interactions. However, most of existing LLM agents choose to sacrifice algorithms' generality by utilizing **environmental tools**, including dynamics (Zhao et al., 2024), available actions (Ding et al., 2023), and reward functions (Hao et al., 2023). For example, Ding et al. (2023) use scripts to determine which actions are available under current state. These methods not only fail to leverage RL for optimal decision-making but also, due to these sacrifices, reduce the generality provided by LLMs.

In this work, we empower the LLM agent with optimal decision-making capability by combining the advantages of the RL and LLM. As shown in Figure 1 (b), we introduce **Model-based LLM Agent with Q-Learning** (MLAQ), a novel LLM agent framework without accessing any environmental tools. The main contribution is three-fold. (1) MLAQ integrates a Q-planner, memory, and imagination space to implement an model-based RL framework in natural language. This allows for generating imaginary transitions with LLMs to minimize environmental interactions, and facilitates Q-learning to optimize the policy for maximizing future rewards. (2) An MCTS-style planning approach is proposed to balance exploration and exploitation within the imagination space. By introducing the concept of virtual nodes, this approach efficiently guides the exploration without any environmental tools, ensuring a sub-linear regret bound by a rigorous theorem. (3) A mixed-examine mechanism is proposed to improve the quality of imaginary transitions. It uses LLM-based self-examine to filter out erroneous transitions in the memory and refines the world model with environmental transitions.

Empirically, we evaluate MLAQ on well-known benchmarks for LLM agents (BlocksWorld (Valmeekam et al., 2022) and RoCo-benchmark (Mandi et al., 2023)), which require optimal decision-making for long horizons. There is no existing LLM agent has successfully obtained the optimal policy, while MLAQ achieves over 90% optimal / success rate across most difficulty levels. The comparison with methods including RoCo (Mandi et al., 2023) and RAP (Hao et al., 2023) fully demonstrates MLAQ's superior performance in optimal decision-making. Through comparative and ablation experiments, we get a key conclusion: integrating the model-based RL framework with an LLM agent in the form of MLAQ can effectively achieve zero-shot optimal decision-making.

## 2 BACKGROUND

**MDP-based framework for an agent.** An MDP can be defined as a tuple $\mathcal{U} = \{\mathbb{S}, \mathbb{A}, \mathbb{T}, \mathbb{R}, \gamma\}$, where $s_t$ is the state in state space $\mathbb{S}$ at timestep $t$, and $a_t$ is the action in action space $\mathbb{A}$, $\mathbb{T}(s_{t+1}|s_t, a_t)$ is the dynamics function, $\mathbb{R}$ is the reward function, and $\gamma$ is the discount factor (Sutton & Barto, 2018). The agent receives a reward $r_t = \mathbb{R}(s_t, a_t, s_{t+1})$ based on changes in state. For single-agent scenarios the agent policy is denoted as $a_t \sim \pi(\cdot|s_t)$. For multi-agent scenarios with agent number $n$, the policy in a centralized manner takes the system as a unified agent to make joint decisions according to global states $\boldsymbol{a}_t = \{a_{i,t}\}_{i=1}^n \sim \pi(\cdot|s_t)$. The policy in a decentralized manner allows each agent $i$ making decisions $a_{i,t} \sim \pi_i(\cdot|o_{i,t}, \rho_{i,t})$ according to its local observation $o_{i,t}$ and communication

message $\rho_{i,t}$, which can be observations (Hu et al., 2023), features (Ding et al., 2024; Chai et al., 2024), or natural language-based dialogues (Mandi et al., 2023; Chen et al., 2024). The state value is defined as the discounted accumulated return under state $s_t$: $V^\pi(s_t) = \mathbb{E}[\sum_{k=0}^{\infty} \gamma^k r_{t+k}]$.

**LLM-based framework for an agent.** Existing approaches construct the above MDP using natural language sentences or paragraphs, benefiting from the autoregressive nature of the Transformer architecture (Vaswani et al., 2017), without the need to align the state and action space dimensions across different domains (Yao et al., 2023a). An LLM-based world model $\hat{\mathbb{T}}(s_{t+1}|s_t, a_t; \tau)$ is utilized to approximate the dynamics function, and an LLM-based basic policy $\pi(\cdot|s_t; \tau)$ is utilized to interact with either the world model or the environment, where $\tau$ is a brief domain description in nature language (Hao et al., 2023). In addition, due to the current limitation of LLMs in accurately evaluating state values, this work adopts a sparse reward setting, meaning that the environment provides a non-zero reward only when the agent reaches a terminal state.

**Q-learning.** Q-learning is a traditional RL algorithm, in which the Q-value indicates the expected return obtained by executing $a$ under $s$. The update rule of the Q function at $k$-th iteration is:

$$Q_{k+1}(s_t, a_t) = Q_k(s_t, a_t) + \alpha \left( r_t + \gamma \max_{a'} Q_k(s_{t+1}, a') - Q_k(s_t, a_t) \right), \qquad (1)$$

where $\alpha$ is the learning rate. As an off-policy approach, Q-learning can use a replay buffer to store transitions $(s, a, r, s')$, enabling policy update through them. With enough iterations and a sufficiently diverse replay buffer, the Q function and its greedy policy can converge to optimality.

**Multi-Armed Bandit.** Given state $s$ with $m$ available actions $\delta(s)$, the successive plays on action $i$ yields (i.i.d.) Q-values, which are sampled from an unknown distribution with an unknown expectation $Q(s, i)$ (Auer et al., 2002). The player aims to minimize the expected regret $\sum_{j:\Delta_j > 0} \Delta_j \mathbb{E}[C_j(N(s))]$, where $\Delta_i = Q^* - Q(s, i)$, $Q^* \doteq \max_i\{Q(s, i)\}$, $N(s)$ is the visit count of $s$, and $C_i(N(s))$ is the selection number of action $i$. The Upper Confidence Bound (UCB) (Auer et al., 2002) could achieve a sub-linear regret bound by selecting actions through the following rule:

$$a^* = \arg\max_{a \in \delta(s)} \text{UCB}(s, a) = \arg\max_{a \in \delta(s)} \left[ V(c(s, a)) + w\sqrt{N(s)/N(c(s, a))} \right], \qquad (2)$$

where $c(s, a)$ is the child node of applying $a$ in $s$, and $w$ is a coefficient.

## 3 METHOD

The essence of MLAQ is to empower LLM agents with optimal decision-making, while also retaining their ability to make decisions with little to no interaction with the environment (i.e., **zero- or few-shot**). While integrating RL-based optimization into LLM agents is relatively straightforward, the key challenge lies in preserving their zero- or few-shot abilities when incorporating RL. In this section, we first illustrate how our agent uses RL approaches to achieve task-specific optimal decision-making under the fully LLM-based framework (Section 3.1). We then introduce how to achieve fully LLM-based imaginary interactions without any environmental tools (Section 3.2). Finally, we present a mixed-examine mechanism to improve the quality of imaginary transitions (Section 3.3).

### 3.1 OPTIMIZING LLM AGENT WITH Q-PLANNER

As shown in Figure 1 (b), we develop an RL-style LLM agent framework, which contains a memory module, a Q-Planner, and an LLM-based imagination space. The RL-based optimization is achieved by Q-Planner, which utilizes the transitions $(s, a, r, s')$ in the memory module to optimize the agent through Q-learning. These transitions are generated from both environmental and imaginary interactions. The overall algorithm is detailed in Appendix F.1.

**Q-Planner.** It serves as the core of the decision-making process, utilizing the optimal decision-making advantage of RL to output the action with the highest Q-value for a given state, where these values are derived from a **tabular** Q-learning iterations in Eq. (1). The tabular Q-function saves the Q values of actions under a state as a list, and outputs the action with the highest Q value when making decisions. We perform Q-learning not directly on the domain-specific memory $\mathcal{M}$ but rather a task-specific replay buffer $\mathcal{D}$ extracted from it, aiming to reduce the exploration space in imaginary interactions. Assuming there are ten available actions, but only two lead to the target state of the current task, exploring the others is completely meaningless.

Figure 2: (a) Imaginary interaction process, which expands the memory and replay buffer through two phases: node selection and transition imagination (might be triggered). (b) Imagine a new transition with self-examine mechanism, where the checkers identify the validity of actions and predictions.

**Replay Buffer.** When a new task starts, a Q function $Q(\cdot, \cdot)$ is obtained by performing Q-learning on $\mathcal{M}$, and the initial replay buffer stores the transitions along the optimal trajectory (if exists) produced by $Q(\cdot, \cdot)$. If this trajectory does not exist or is demonstrated to be non-optimal through environmental interactions, the agent keeps exploring in the imagination space to expand the replay buffer until the best trajectory is demonstrated to be optimal. This process can be seen in Appendix F.

**Imaginary Interaction.** As shown in Figure 1 (b), an LLM-based basic policy and world model are established to perform imaginary interactions. Without accessing the environment, their understanding of the domain is entirely reliant on the given domain description (manual). Therefore, MLAQ could preserve the zero-shot decision-making capabilities of LLM agents while satisfying the training data requirements of Q-learning. The details will be described in the subsequent section.

**Environmental Interaction.** MLAQ agent obtains the current state and domain description $\tau$ from the environment, explores the imagination space to derive the optimal policy using the Q-planner, and then outputs actions to interact with the environment. The domain description $\tau$ includes the domain goal, state/action definitions, and also some detailed constraints for decision-making and predicting. Please refer to Appendix J for more detail.

## 3.2 LLM-BASED IMAGINATION FOR MLAQ

This section presents an MCTS-style planning method that balances exploration and exploitation while using only LLMs to generate imaginary transitions, thus efficiently expanding the memory and replay buffer. Without any environmental tools, MLAQ should the available actions for any seen states from scratch. Given a new task, MLAQ performs multiple **imagination rounds**, with each consisting of an imaginary trajectories from the original state to the target state. An imagination round comprises two phases: node selection and transition imagination. See Appendix F.2 for pseudo-codes.

**Node selection.** This phase involves balancing exploration and exploitation through a novel selection rule. As denoted in Figure 2 (a), the transitions $(s, a, r, s')$ in the replay buffer adopt a tree-like structure, wherein two nodes $s \rightarrow s'$ are linked by an action and reward edge $(a, r)$. The original UCB treats planning as a multi-armed bandit problem, but using LLMs for planning without environmental tools presents **a variation**, as the ground-truth available actions $\delta(s)$ are unknown. Therefore, we begin with an empty available action set $\hat{\delta}(s)$ and introduce the concept of virtual nodes to gradually expand it. Within the replay buffer, a actual (state) node is created based on a state $s$, and a virtual node is appended as its first child node, with the edge is a virtual action. Then, MLAQ starts with the empty $\hat{\delta}(s)$ and iteratively leverages LLMs to explore potential actions and expand it. This process can be modeled as follows:

*A **Variant of Multi-Armed Bandit.** In a scenario with $m$ available actions ($a = \{1, 2, \ldots, m\}$) in state $s$, initially only the virtual node $y$ can be selected. Selecting $y$ enables the least-indexed non-selectable node in $a$ to be selectable, while selecting other node involves estimating its Q-value.*

When $y$ is selected for the $j$-th times, i.e., $|\hat{\delta}(s)| = j$, the actual nodes $\{1, 2, \ldots, j\}$ can be selected by the planning policy. Moreover, due to the introduction of virtual nodes, we propose a variant of the original UCB algorithm by modifying its selection rule as follows:

$$a^* = \arg\max_{a \in \hat{\delta}(s)} \mathrm{vUCB}(s, a) = \arg\max_{a \in \hat{\delta}(s)} \left[ V\big(c(s, a)\big) + f\big(N(s), N(c(s, a))\big) \right], \quad (3)$$

where $f(N(s), N(c))$ is the confidence bound, defined as $w\sqrt{N(s_t)/N(c(s_t, a))}$ for actual nodes and $g(|\hat{\delta}(s_t)|)\sqrt{N(s_t)/|\hat{\delta}(s_t)|}$ for virtual nodes. $g(x)$ is a coefficient function that must satisfy some properties, which are discussed in the Appendix C. However, since we do not know the number of available actions, it is set to $g(x) = w_g(e^{-x^2} - \epsilon_g)$ in practice, where $\epsilon_g$ and $w_g$ are hyper-parameters.

**Theorem 1.** *Suppose the player employs the node selection rule defined in (2). Then, in a scenario with $m$ available actions in state $s$, the bound of the expected cumulative regret is:*

$$\text{Regret} = \sum_{j:\Delta_j > 0} \Delta_j \mathbb{E}[C_j(N(s))] \le [8\sum_{i:\Delta_i > 0}(\frac{\ln N(s)}{\Delta_i})] + (1 + \frac{\pi^2}{3})(\sum_{i=1}^{m}\Delta_i), \quad (4)$$

*where $\Delta_i = Q^* - Q(s, i)$ for action index $i$ in $\{1, 2, ..., m\}$ and $Q^* \doteq \max_i\{Q(s, i)\}$, $N(s)$ is the visit count of $s$, and $C_i(N(s))$ is the selection number of action $i$.*

This theorem demonstrates that our proposed variant of UCB shares the same sub-linear regret bound, enabling efficient exploration and exploitation without environmental tools. Please refer to the Appendix C for the proof. MLAQ turns to the **transition imagination** phase if virtual nodes are selected. Otherwise, it iterates until a terminal state is reached. A terminal state denotes either success states when reaching target states or failure states when the LLM-based basic policy determines no actions are available. When a terminal state is reached, an imagination round ends.

**Transition imagination.** This phase is triggered when selecting virtual nodes and generates an imaginary transition. Beginning from $s_t$, we first mark the existing actions in $\hat{\delta}(s_t)$ as forbidden, where the specific prompt can be found in Appendix G. We then instruct the basic policy to output an action $a_t$ excluding forbidden actions to explore and expand $\hat{\delta}(s_t)$. If the pair $(s_t, a_t)$ could be queried in $\mathcal{M}$, we directly **re-utilize** the stored next state $c(s_t, a_t)$ as its child state to skip LLM queries. Otherwise, we employ the LLM-based world model to predict the next state $s_{t+1} \sim \hat{\mathbb{T}}(s_t, a_t; \tau)$. The reward is also calculated by LLMs, following a sparse reward setting. In addition, if the basic policy determines that no available action exists except forbidden actions, we select the best action in $\delta(s)$ to continue the imaginary interaction. Since we **cannot confirm** if the available actions are exhausted, the virtual node always remains after the addition of the next state node.

### 3.3 MIXED-EXAMINATION FOR IMAGINARY TRANSITIONS

Within the imagination space, we propose a mixed-examination mechanism to improve the quality of imaginary transitions without accessing any environmental tools.

**Env-examination.** MLAQ treats environmental transitions as ground truth to correct the imaginary transitions and refine the LLM-based world model. When using domain descriptions to understand the environment for LLMs, both LLM's comprehensions and descriptions could be incomplete, leading to a gap between the LLM-based world model and the environment dynamics. To address this, during interactions with the environment, MLAQ corrects the transitions stored in memory based on environmental transitions and records the discrepancies into the world model's prompt for refinement.

**Self-examination.** Considering the **hallucinations** inherent in LLMs, even when the prompt contains complete information about the environment, the model may still produce incorrect outputs. To this end, as shown in Figure 2 (b), MLAQ adopts an LLM-based action checker and prediction checker to check the validity of the outputs of the basic policy and world model. Current models (e.g., GPT-4) already possess the ability to improve output accuracy through self-examination, which will be validated in subsequent experiments. Prompt templates of the two checkers are provided in Appendix G. Take the action checker as an example, given the current state and the action should be checked, it queries the LLM to check if the action breaks any constraint following the instructions. The checkers continuously check and provide feedback until the outputs are correct.

## 4 EXPERIMENTS

In this section, we conduct experiments in several challenging environments for LLM agents to answer the following questions: **(A)** *Does the introduction of a complete RL framework improve the optimal decision-making capabilities of LLM agents?* **(B)** *Can the MLAQ agent preserve zero- or*

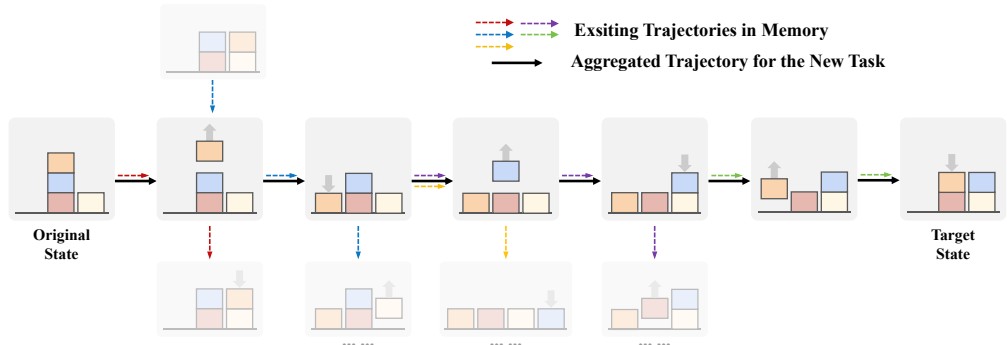

Figure 3: An instance with an optimal step of 6. It illustrates that MLAQ, due to its ability to leverage transitions across multiple tasks within the same domain, can obtain an optimal policy through RL-based optimization without requiring any search in the imagination space when given a new task.

*few-shot decision-making capabilities?* **(C)** *Does the current LLM have the potential to improve its output performance through self-examination?* **(D)** *Does expanding the memory scope from task-specific to domain-specific improve the agent performance?*

## 4.1 EXPERIMENTAL SETUP

We conduct experiments on the BlocksWorld benchmark (Valmeekam et al., 2022) for the single-agent setting and the RoCo-benchmark (Mandi et al., 2023) for the multi-agent setting. Agents in these domains require multi-step decision-making to achieve the final goal, necessitating the ability to maximize expected future rewards. Additionally, the decision-making space for LLM agents in the RoCo-benchmark is significantly larger than that in BlocksWorld due to the presence of multiple agents. The details of these benchmarks can be found in Appendix D.

We compare MLAQ with CoT (Wei et al., 2022), RAP (Hao et al., 2023), Rex (Murthy et al., 2023), RAFA (Liu et al., 2023), and RoCo (Mandi et al., 2023). In line with RAP (Hao et al., 2023), we group all tasks by their **optimal steps**, indicating the length of the tasks' optimal decision sequences. Within a domain, the experiments are conducted from the tasks with smaller optimal steps to those with larger steps. Consequently, only the memory size of MLAQ would increase gradually with the experiments due to its domain scope, empowering the LLM agent with optimal decision-making capability with Q-Planner. Furthermore, we also conduct exhaustive ablation experiments to verify the effect of our domain-specific memory module, further expansion in memory, and the mixed-examination mechanism. The details of hyper-parameters can be found in Appendix I.

## 4.2 MAIN RESULTS

**Single-agent scenario.** As shown in Table 1, the optimal steps of BlocksWorld vary from 2 to 12. The **optimal rate** in this table denotes that the agent could reach the target state under the optimal steps. Our agent achieves more than 90% optimal rate across all difficulty levels, indicating an affirmative response to **Question A**. In contrast, the other LLM agent can only succeed in tasks with smaller optimal steps. In order to qualitatively provide an affirmative answer to **Question D**, we use a task with 6 optimal steps in Figure 3 as an example to backtrack the transitions in memory. In this case, MLAQ agent finds an optimal decision sequence, represented by black solid arrows,

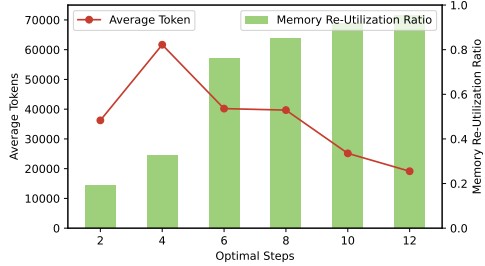

Figure 4: Tendency of memory re-utilization ratio and token consumption.

from the initial replay buffer **without** querying LLMs. The colored dashed arrows indicate existing trajectories stored in memory. This 6-step sequence is aggregated from trajectories of multiple tasks, including two 2-step and three 4-step trajectories. This clearly demonstrates the effectiveness of

Table 1: The optimal rate of experimental methods in BlocksWorld domain.

| Methods | 2-step | 4-step | 6-step | 8-step | 10-step | 12-step |
|---|---|---|---|---|---|---|
| CoT (Wei et al., 2022) | 0.22 | 0.14 | 0.02 | 0.02 | 0.00 | 0.00 |
| REX (Murthy et al., 2023) | 0.80 | 0.45 | 0.25 | - | - | - |
| RAFA (Liu et al., 2023) | - | 0.97 | 0.75 | - | - | - |
| RAP (Hao et al., 2023) | 0.67 | 0.76 | 0.74 | 0.48 | 0.17 | 0.09 |
| MLAQ | **1.00** | **1.00** | **1.00** | **0.97** | **0.93** | **0.90** |

our domain memory in enhancing the agents' decision-making capabilities in long-horizon tasks. Quantitative analysis will be provided in the ablation experiments.

Besides, we also investigate the memory re-utilization ratio in transition imagination phase. As shown in Figure 4, higher re-utilization ratio commonly leads to a lower average consumed tokens. As the increase of the optimal step, the ratio of memory re-utilization increases from 19% to 90%. Meanwhile, the average token consumption of each task first increases and then decreases significantly. The increase of tokens from 2-step task to 4-step task is attributed to the increased task difficulty, requiring the agent to employ more imagination to make optimal decisions. Then, as the memory becomes enriched with the data collected from resolved tasks within the same domain, the probability of re-utilizing stored transitions increases, leading to a reduction in average token consumption.

**Multi-agent scenario.** In RoCo-benchmark, robotic arms need to efficiently allocate tasks internally and collaborate to achieve the final goal. Table 2 shows the experimental results of RoCo and MLAQ on Sort domain. MLAQ achieves a success rate approaching 100% across all difficulty levels and maintains an optimal performance exceeding 50% across most tasks, further giving an affirmative response to **Question A**. In this table, we introduce "success rate" as an additional metrics as the expanded search space from the presence of multiple agents significantly reduces their optimal rate. The notation "MLAQ⁻" denotes that MLAQ agent terminates its imagination upon reaching the target state, without further exploring to get a better policy. A detailed analysis of "MLAQ⁻" is provided in the ablation section. The optimal steps for Sort range from 1 to 6, with MLAQ consistently achieving higher success and optimal rates than RoCo, especially in long-horizon tasks. Due to space limitations, the results and analysis on Sandwich domain are provided in Appendix H.2.

Table 2: Comparison of multiple metrics between MLAQ and RoCo in Sort domain.

| Metrics | Methods | 1-step | 2-step | 3-step | 4-step | 5-step | 6-step | Average |
|---|---|---|---|---|---|---|---|---|
| Success Rate | RoCo | 1.00 | 0.64 | 0.47 | 0.10 | 0.03 | 0.00 | 0.35 |
| | MLAQ | **1.00** | **0.96** | **0.97** | **1.00** | **0.93** | **1.00** | **0.98** |
| Env Replans (n-shot) | RoCo | 0.30 | 6.30 | 5.60 | 9.74 | 7.67 | 6.92 | 6.41 |
| | MLAQ | **0.00** | **0.04** | **0.03** | **0.10** | **0.10** | **0.04** | **0.06** |
| Optimal Rate | RoCo | 0.80 | 0.36 | 0.27 | 0.00 | 0.00 | 0.00 | 0.21 |
| | MLAQ⁻ | 0.95 | 0.64 | 0.57 | 0.67 | 0.33 | 0.43 | 0.58 |
| | MLAQ | **1.00** | **0.86** | **0.77** | **0.73** | **0.50** | **0.75** | **0.75** |
| Average Token | RoCo | 10605 | 530817 | 332730 | 305175 | 345762 | 320045 | 322630 |
| | MLAQ⁻ | **7093** | **15104** | **16436** | **22197** | **18133** | **8220** | **15216** |
| | MLAQ | 8491 | 66367 | 156490 | 119916 | 409495 | 243151 | 175560 |
| Optimal Gap | RoCo | 0.35 | 3.37 | 3.43 | 3.33 | 2.97 | 2.00 | 2.72 |
| | MLAQ⁻ | 0.10 | 0.64 | 0.73 | 0.53 | 1.20 | 0.46 | 0.65 |
| | MLAQ | **0.00** | **0.32** | **0.40** | **0.43** | **0.80** | **0.25** | **0.39** |
| Memory Re-Util. Ratio | | 0.27 | 0.61 | 0.71 | 0.72 | 0.73 | 0.83 | 0.66 |

In Table 2, the **average token** consumption of each task still shows a trend of increasing first and then decreasing, and is lower than RoCo's in Sort. The **Env Replans** metric denotes the frequency of environment feedback, which reflects the **zero-shot** decision-making ability of the LLM agent. A smaller Env Replans indicates that the agent requires less ground-truth information from the environment to refine its decisions. The results show that the number of environmental

Table 3: The optimal rate for optimizing multiple epochs in Sort domain.

| Methods | 1-step | 2-step | 3-step | 4-step | 5-step | 6-step |
|---|---|---|---|---|---|---|
| MLAQ-1st epoch | **1.00** | 0.86 | 0.77 | 0.73 | 0.50 | 0.75 |
| MLAQ-2nd epoch | **1.00** | **0.93** | **0.90** | **0.77** | **0.57** | 0.88 |
| MLAQ-3rd epoch | **1.00** | **0.93** | **0.90** | **0.77** | **0.57** | **0.92** |

replans for MLAQ has decreased by approximately two orders of magnitude compared to RoCo, indicating that the MLAQ agent almost does not require environmental transitions to correct imaginary transitions. In contrast, existing methods require the environment to verify action's validity and provide environmental feedback, facilitating LLM agents to replan until actions are available (Mandi et al., 2023; Shinn et al., 2023). This result indicates an affirmative answer to the **Question B**. In addition, since we provide two types of basic policy (central and dialogue) in multi-agent scenarios, we also conduct experiments to compare them in Appendix H.1.

## 4.3 MLAQ WITH MULTIPLE EPOCHS

This experiment aims to demonstrate the potential of MLAQ to enhance its optimal decision-making ability through iterative optimization. In previous experiments, each domain's experiment starts with an empty memory. In this section, we conduct multiple epochs of MLAQ experiments by performing MLAQ with a non-empty memory to further enhance its decision-making capabilities and utilize memory re-utilization to reduce token consumption. At the beginning of each epoch, the input memory is the memory of the last MLAQ epoch after finishing all given tasks of the domain. As shown in Table 3 and Figure 5, an additional training epoch results in an improvement of the optimal rate while reducing token consumption. Although MLAQ with multiple epochs no longer constitutes zero-shot decision-making, the small magnitude of env replans implies that minimal environmental information is injected into subsequent epoch decisions. Consequently, it still provides a more definitive affirmative answer to **Question A** and **Question B**.

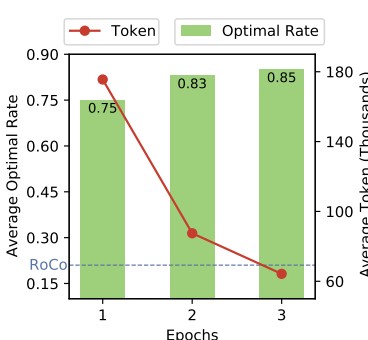

Figure 5: Tendency of optimal rates and tokens with epochs.

## 4.4 ABLATION RESULTS

The key components of MLAQ are the Q-planner, memory, and imagination space. In this section, we validate their effects by individually diminishing their capabilities. The ablation experiments are conducted under 8-step tasks in the BlocksWorld domain. This is primarily because the task difficulty at this optimal step is moderate for an appropriate comparison.

Table 4: Ablation results in BlocksWorld domain.

| Ablation Methods | Optimal Rate | Token | Env Replans |
|---|---|---|---|
| RAP | 0.48 | - | - |
| MLAQ w/o domain memory | 0.87 | 378941 | 0.02 |
| MLAQ w/o self-examine | 0.57 | 20115 | 0.81 |
| MLAQ w/o env-examine | 0.95 | 40772 | 0.05 |
| MLAQ | 1.00 | 39688 | 0.02 |

**Removing further optimization.** This experiment evaluates the impact of the Q-planner's inability to further optimize. In our framework, if the best decision sequence derived from the Q-Planner is determined to non-optimal through environmental interactions, the MLAQ agent would keep exploring in the imagination space to continually expand the memory and replay buffer for further optimization. Within the imagination space, we allow the MLAQ agent to complete only one successful imagination round by terminating the exploration process upon reaching the target state, referring to this method as MLAQ$^-$. As demonstrated in Tables 2, this deletion of further optimization

decreases token consumption while concurrently diminishing the optimal rate. The 'Optimal Gap (OG)' represents the gap between average completion steps and optimal steps. The results indicate that MLAQ achieves lower OG than MLAQ$^-$, which fully demonstrates the effectiveness of the expansion of the replay buffer and the optimization based on Q-Planner. The experiments on the Sandwich domain presented in Appendix H.2 also yields the similar results.

**Removing LLM-based self-examine and env-examine.** This experiment reduces the quality of transitions generated in the imaginative space by separately removing self-examine and env-examine, thereby evaluating their effect. (1) The **self-examine** utilizes an LLM-based action checker and prediction checker to enable error correction within the imagination space. We remove it from MLAQ agent to directly output the actions and predictions without LLM-based examination, which is denoted as **MLAQ w/o self-examine** in Table 4. Note that the initial memory stores transitions from the 2-step to 6-step experiments rather than starting from scratch. The results demonstrate a notable reduction in token consumption alongside a significant decrease in the optimal rate. Moreover, the frequency of environmental feedback has significantly increased, providing support for affirmative answers to **Question C**. (2) The **env-examine** corrects the imagination process by treating environmental transitions as ground truth. Removing it allows for an evaluation of the imagination capability using only the LLMs. The results of **MLAQ w/o env-examine** indicate only a slight performance decline compared to MLAQ. Combined with the env replans metrics of MLAQ, this leads to the conclusion that the current LLMs (e.g., GPT-4) possess the capability to efficiently generate imaginary transitions through self-examination, resulting in a relatively low dependency on ground truth, answering **Question C** from the opposing viewpoint. Removing both of them will lead to a decline in the quality of imaginary transitions, thereby reducing the decision-making performance of our MLAQ agent.

**Narrowing memory scope.** This experiment narrows the scope of the memory module from domain-specific to task-specific like existing methods (Hao et al., 2023) in order to evaluate the effect of constructing memory in an RL manner. This approach is denoted as **MLAQ w/o domain memory** in Table 4. The results indicate a significant increase in token consumption alongside a decrease in the agent's optimal rate, quantitatively answering the **Question C**. This decline stems from the MLAQ agent's inability to leverage prior decision-making experiences to solve new tasks within the same domain. However, the optimal rate of the MLAQ agent still exceeds that of RAP, owing to the influence of other components within the MLAQ framework.

## 4.5 SELF-EXAMINATION ANALYSIS

This section quantitatively evaluates the LLM-based self-examination, showing that while hallucinations may occur during the examining process, performing multiple checks in transition imagination significantly reduces the erroneous transitions. In the self-examination process, the LLM-based checkers may erroneously identify correct outputs as incorrect, and vice versa. We sample 128 random states from the Sort domain, providing correct actions for 64 states and incorrect actions for the others, with each incorrect action breaking only one constraint. The testing for the prediction follows a similar approach, where each incorrect prediction only violates one constraint.

Table 5: Results on identification capability of the checkers.

| Checker Type | TP $\uparrow$ | FN $\downarrow$ | FP $\downarrow$ | TN $\uparrow$ | Precision $\uparrow$ |
|---|---|---|---|---|---|
| Action Checker | 63 | 1 | 6 | 58 | 91.3% |
| Prediction Checker | 64 | 0 | 10 | 54 | 86.5% |

As shown in Table 5, we evaluate the performance of the checkers using five metrics. **True Positives (TP)** identify correct outputs (actions or predictions) as correct, **False Negatives (FN)** identify correct outputs as incorrect, **False Positives (FP)** identify incorrect outputs as correct, **True Negatives (TN)** identify incorrect outputs as incorrect, and **Precision = TP / (TP + FP)** is a proportion of correctly identified correct cases among all cases identified as correct. Results in Table 5 suggest that while both checkers accurately identify correct outputs, they may misidentify incorrect outputs as correct ones, leading to inaccuracies in imaginary transitions and subsequent impacts on agent performance.

To further investigate the impact of inaccuracies in self-examination, similar methods are used to test the accuracy of the basic policy and world model. Without forbidden actions, the accuracy of the basic policy is 84.4%, while the accuracy of the world model is 100%. Therefore, the combined

calculation yields a probability of 1.36% = (1 - 84.4%) * [6 / (6 + 63)] for an wrong transition to be stored in memory. Compared to the probability of 15.6% = (1 - 84.4%) without self-examination, MLAQ can greatly improve the quality of imaginary transitions. These results provide support for affirmative answers to **Question C**. Furthermore, some erroneous transitions stored in memory could be excluded from the best policy derived from RL-based optimization due to their low values, while the others could be corrected by env-examination. Experimental data in Table 2 also indicate that these errors minimally impact the decision-making capabilities of LLM agents.

## 5 RELATION TO OTHER METHODS

In this section, we compare the MLAQ framework with several representative works in LLM agents. Table 6 presents the interaction between Agent, World Model (WM), and Environment (Env) in existing approaches, as well as the forms of their memory. (1) The **Agent-Env** interaction is central to most LLM agents, with efforts to optimize agents through techniques like self-reflect (Shinn et al., 2023) and dialogue (Mandi et al., 2023) for prompt enhancing, and Monte Carlo Tree Search (MCTS) for transition-level optimization (Ding et al., 2023). (2) The **Agent-WM** interaction is originated from model-based RL (Hafner et al., 2020) to expand memory through the agent's imaginary interactions. RAP (Hao et al., 2023) and ToT (Yao et al., 2023a) construct an LLM-based world model for MCTS-based planning in a single task, but they still not try to narrow the gap between the world model and environment. (3) The **WM-Env** interaction requires using the environmental data to correct the mistakes made by LLM-based world model, which is achieved by LATS (Zhou et al., 2023a). (4) Finally, the memory modules of LLM agents differ in the **level and scope**. Insight-level memory stores agents' natural language summaries of their insights (Ding et al., 2023), while transition-level memory retains RL-like transitions (Hao et al., 2023). The scope here represents the time span of the data in memories, encompassing (a) Step: the single transition, (b) Task: multiple transitions within the same task, and (c) Domain: multiple transitions in multiple tasks within the same domain. We present different scopes of memory in Appendix A, and detailed related works in Appendix B.

Table 6: Different interactions in existing LLM-based decision-making methods.

| Method | Agent-Env | Agent-WM | WM-Env | Memory Level-Scope |
|---|---|---|---|---|
| Reflexion (Shinn et al., 2023) | Self-reflection | - | - | Insight-Task |
| Expel (Zhao et al., 2024) | Self-reflection | - | - | Insight-Domain |
| RoCo (Mandi et al., 2023) | Dialog Feedback | - | - | Transition-Step |
| XoT (Ding et al., 2023) | MCTS Extraction | - | - | Transition-Task |
| RAP (Hao et al., 2023) | Direct | MCTS | - | Transition-Task |
| ToT (Yao et al., 2023a) | Direct | MCTS | - | Transition-Task |
| LATS (Zhou et al., 2023a) | Self-reflection | MCTS | Env-examine | Transition-Task |
| MLAQ (Ours) | Q-Planner | UCB-guided | Mixed-examine | Transition-Domain |

## 6 CONCLUSION

This work introduces MLAQ, a powerful MDP-based LLM agent framework that achieves zero-shot optimal decision-making. MLAQ consists of a memory module, a Q-planner, and an imagination space to fully leverage the general comprehension capabilities of LLMs and the optimization capabilities of RL. The imagination space is entirely based on LLM to generate imaginary transitions, which are then provided to the Q-planner for RL-based optimization. The proposed planning algorithm balances the exploration and exploitation by introducing the concept of virtual nodes, while ensuring a sub-linear regret bound guaranteed by a theorem. Moreover, a mixed-examine mechanism is employed to improve the quality of the imaginary transitions. We evaluate MLAQ on both single-agent BlocksWorld and multi-agent RoCo-benchmark domains, comparing it with several advanced LLM agents. Results indicate that MLAQ outperforms existing methods, especially in tasks involving long-horizon decision-making. Furthermore, we conduct exhaustive additional experiments to investigate how MLAQ enables LLM agents to achieve zero-shot optimal decision-making capabilities. The core conclusion of this work is that constructing a model-based RL framework in natural language for current LLMs can further enhance the decision-making abilities of LLM agents. In future work, we intend to deploy this method on physical robotic platforms to fully exploit its advantages.

## ACKNOWLEDGMENTS

This work was supported in part by the National Natural Science Foundation of China under Grant 62136008 and Grant 62293541, in part by Beijing Natural Science Foundation under Grant 4232056, and in part by Beijing Nova Program under Grant 20240484514.

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

## A    DETAILED DESCRIPTIONS OF MEMORY SCOPE

MLAQ employs a memory module with domain scope. In order to better illustrate its difference from the memory scope in previous LLM agents, we present the memory module under different scopes in Figure 6. Figure 6 (a) shows a memory with **Step** scope, which only retains the interaction data between the agent and the environment at the current time step. For instance, when refining LLM agent's decisions, RoCo (Mandi et al., 2023) only relies on the feedback provided by the environment in the current state. Figure 6 (b) shows a memory with **Task** scope, which focuses on solving the current given task and retains the data obtained from the interactions between the agent and the environment in this task. Most LLM agents adopt this form of memory, which is called a **replay buffer** in Figure 1. Take the RAP (Hao et al., 2023) agent as an example, it explores within an LLM-based imagination space and retains visited trajectories to avoid repeated LLM-based state predictions. However, the memory for each task is unique and contains only information relevant to this single task, meaning that the agent cannot gain experience from the decision histories of other tasks within the same domain to solve the current task. Instead, our memory with domain scope stores environmental and imaginary interaction data from all tasks in the domain, effectively reducing the need for LLM queries and improving the efficiency of the reinforcement learning through Q-planner. Besides, one point to **clarify** is that in Figure 6 (c), cylinders are depicted in different colors only to illustrate that they have transitions from different tasks. The transitions stored in our memory do not contain any task-specific labeling, and are always stored in the format of $(s, a, r, s')$.

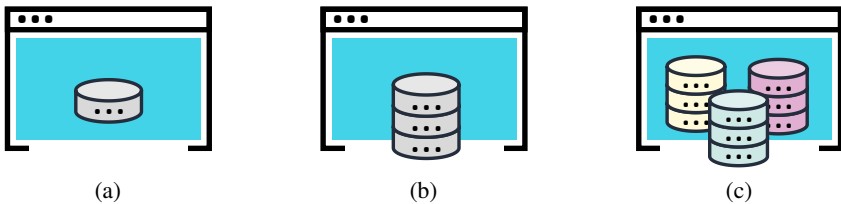

(a)                                     (b)                                     (c)

Figure 6: Memory modules with different scope. (a) Step. (b) Task. (c) Domain.

## B    RELATED WORK

Leveraging the Transformer network architecture, recent advancements in Large Language Models (LLMs) have highlighted their potential in decision-making tasks. These approaches are commonly referred as LLM agents. The research on LLM agents originated from GPT-3, an LLM with 175 billion parameters (Brown et al., 2020). Its authors found that the inference and decision-making ability of LLM can be improved by simply adjusting the input prompt without adjusting the model parameters. This insight leads to the development of the Chain-of-Thought (CoT) approach, which improves the LLM's reasoning capabilities by integrating step-by-step reasoning examples into the prompts (Wei et al., 2022). Subsequent studies have focused on further enhancing LLM reasoning capability through prompt engineering. Techniques such as task decomposition, exemplified by least-to-most (Zhou et al., 2023b) and DecomP (Khot et al., 2022) prompting, are proposed to simplify the task complexity. The aforementioned methods also could address some traditional problems in natural language processing, such as cloze and completion task and reasoning task (Turpin et al., 2023). With the development of LLMs, OpenAI's O1 has been characterized as a Large Reasoning Model (LRM), demonstrating preliminary planning capabilities (a success rate of 97.8% in BlocksWorld as reported by Valmeekam et al. (2024)). However, due to our experiments employ a more challenging **optimal rate** as the metric, the direct comparison is not presented in the main text.

**LLM agents with prompt optimization.** As research progresses, an increasing number of studies focus on developing an LLM agent to tackle decision-making tasks with higher complexity. Most methods adopt the existing paradigm, aiming to indirectly enhance the decision-making capabilities of LLM agents by optimizing the prompts given to LLMs (Brown et al., 2020). These approaches typically involve enriching the prompt's informational content through environmental or human feedback and employing some techniques to summarize these information into some insights, thereby minimizing the prompt's length (Zhang et al., 2024d). Huang et al. (2022b) enable the LLM agent

to understand many semantic aspects of the world by leveraging environmental feedbacks to form an inner monologue, acknowledging what, when, and how to do skills in embodied environments. DEPS (Wang et al., 2023b) involves the description and self-explanation of the agent's plans to improve the accuracy of multi-step reasoning in long-horizon tasks. It also provides a trainable goal selector to rank the sub-goals based on the estimated steps of completion, thus refining the agent's plans. Furthermore, as shown in Table 6, the self-reflect is a widely-used technique to improve LLM agent's decision-making capability. Reflexion (Shinn et al., 2023) and PlanAgent Zheng et al. (2024) empowers autonomous LLM agents to iteratively revise their actions based on feedback, thereby optimizing their behavior. ReAct (Yao et al., 2023b) introduces a closed-loop reasoning process by alternately generating verbal reasoning traces and actions to achieve self-correction of knowledge. Additionally, it acquires additional information from external sources through the execution of actions. AgentVerse (Chen et al., 2024) involves multiple LLM agents with different roles to orchestrate a collaborative group of expert agents. These agents evaluate their executed actions to adjusting the group and discuss collaboratively for making better decisions. There are also numerous approaches aimed at addressing decision-making problems in multi-agent environments. ProAgent (Zhang et al., 2024a) considers scenarios involving cooperation with human players. It performs cooperative reasoning and planning based on inference of teammates' behavior, corrects the agents' beliefs using actual decisions, and stores acquired knowledge into a task memory. Mandi et al. (2023) proposes a multi-agent benchmark RoCo-benchmark with multiple robotic arms solving a cooperative task. It also constructs a dialog-based multi-agent framework RoCo to address the proposed benchmark. CoELA (Zhang et al., 2024b) aims to address the multi-agent cooperation problem with decentralized control by providing communication between LLM agents to exchange their information and knowledge. Its framework extensively utilizes the potential of LLM in perception, communication, planning, memory and execution.

**LLM agents with action optimization under MDP framework.** Those methods, which only aim at optimizing prompts, cannot achieve satisfactory performance in long-horizon decision-making tasks (Hao et al., 2023), which are more challenging to LLM agents. Firstly, the inherent reasoning capabilities of these methods, bounded by the fixed parameters of the base LLM, fall short in complex tasks necessitating long-horizon decision sequences. Secondly, adjusting input prompts may improve reasoning to some extent but does not fundamentally optimize the policy of the LLM agent for acquiring **optimal** decision sequences. Recent researches have explored using broader language applications to model the environmental dynamics and reward functions, employing planning algorithms to guide decision-making (Liu et al., 2023; Zhang et al., 2024d). They formulate the LLM agent under the Markov Decision Process (MDP) framework by decomposing original long-horizon decision sequences into atomized state transitions. Yao et al. (2023a) propose Tree-of-Thoughts (ToT), which establishes an incomplete MDP framework limited to state space, action space, and value function components. ToT employs depth-first and breadth-first search techniques to derive optimal sequences of thoughts in the decomposed "thought space". RAP (Hao et al., 2023) leverages LLMs to explicitly construct a world model, serving as the dynamics function within the MDP. RAP employs an LLM-based policy to generate decision trajectories by interacting with the world model within the imagination space and utilizes the MCTS algorithm to effectively balance exploration and exploitation, thus optimizing decision sequences. REX (Murthy et al., 2023) introduces an additional layer of rewards and incorporates principles akin to Upper Confidence Bound (UCB) values to further enrich exploration within the action space, resulting in more robust and efficient agent performance. Everything-of-thought (XoT) (Ding et al., 2023) emphasizes the "Penrose triangle" in traditional LLM agents, indicating that Performance, Efficiency, and Flexibility cannot simultaneously exist. It utilizes LLM for thought generation and employs RL and MCTS algorithms to revise the generated thought tree, thereby enhancing the agent's performance. Zhao et al. (2023) establish an LLM-based commonsense world model and basic policy to conduct MCTS-based searches. External knowledge is obtained through datasets or interactions with human players, enabling updates to the belief of environmental states and consequently making optimal decisions. Kong et al. (2024) integrates an API Retriever, LLM Finetuner, and Demo Selector within a unified framework to tackle the challenges of task planning and tool usage in complex real-world systems. Expel (Zhao et al., 2024) maintains an insight-level memory for the LLM agent. After interacting with the environment, the agent learns from successful trajectories and avoids failed trajectories to summarize a series of insights. These insights are used to enrich the LLM agent's prompts, thereby enhancing its decision-making capabilities. However, most of them require the environmental functions to enhance decision-making and lack the capability to leverage experiences from other tasks within the same domain.

There are also some LLM agents aiming to be a zero-shot planner. Kwon et al. (2024) design many task-agnostic prompts to investigate which design choices in this prompt are the most important, thus developing a zero-shot robot manipulator. Huang et al. (2022a) decompose high-level tasks into mid-level plans without any further training of pre-trained LLMs, and propose a procedure to semantically translate the plans to available actions in a zero-shot manner. PS Prompting improves upon zero-shot-CoT at the prompt level, enabling LLMs to implement planning capabilities (Wang et al., 2023a). Gkanatsios et al. (2023) present an energy-based framework that converts language instructions into optimizable functions to guide object rearrangement through visual-motor policies. However, these approaches mostly design prompts for specific domains and still cannot achieve optimal decision-making.

**RL approaches in agents.** Traditional RL and MARL approaches require RL-based optimization techniques to update the parameters of agent policies. These methods either collect interaction data between agents and the environment (Rashid et al., 2018; Chai et al., 2021; Hu et al., 2023) or use supervised learning to construct a world model that serves as a digital replica of the real environment (Chai et al., 2024; Egorov & Shpilman, 2022), providing imagined interaction data (Hafner et al., 2020). In contrast, the training of LLMs involves a vast and diverse dataset, enabling the construction of task-specific basic policies and world models through task descriptions in natural language (Hao et al., 2023; Yao et al., 2023a), thereby eliminating the need for parameter updates of traditional methods. Reflextion (Shinn et al., 2023) maintains an RL-like framework, which includes Actor and Critic modules used for action output and action evaluation, respectively. However, it does not employ RL-based optimization at the transition level based on the Bellman equation. Instead, it heavily relies on the decision-making capabilities of the LLM itself, leading to poor performance in long-horizon tasks. Zhang et al. (2024c) utilize RL-based critic regression to learn a sequential advantage function and treat the LLM planner as an optimizer to generate actions that maximize this function. Xu et al. (2023) propose an LLM agent for Werewolf game, which utilizes LLM to generate candidate actions in the current state, and then uses a **one-step** population-based RL training process to select the optimal action from these actions. MCTS is also a traditional RL approach, but it requires prior knowledge about environmental functions and focuses more on the search process. Some of the previously introduced methods (Hao et al., 2023; Zhao et al., 2023) use MCTS for planning and achieve optimal decision-making based on some strong assumptions, but still do not perform satisfactory performance in long-horizon tasks.

## C   PROOF FOR THEOREM 1

Before presenting the proof of Theorem 1, we first introduce Theorem 2 from (Auer et al., 2002) and provide part of the derivation to facilitate the subsequent steps. It is important to note that some variables in this section may cause confusion with similarly defined variables in the main text. These variables are introduced temporarily to improve the readability of the derivation and are only valid within the proof of this section. For instance, while $t$ denotes the time in the main text, it will be used here to denote the number of UCB rounds.

**Theorem 2.** *For all $m > 1$, if policy UCB is run on state $s$ with $m$ available action, performing each action yields arbitrary Q-value distributions $P_1, ..., P_m$ with support in $[0, 1]$, then its expected regret after any round number of $T$ is at most:*

$$[8 \sum_{i:\Delta_i>0} (\frac{\ln N(s)}{\Delta_i})] + (1 + \frac{\pi^2}{3})(\sum_{i=1}^{m} \Delta_i), \tag{5}$$

*where $Q(s, 1), ..., Q(s, m)$ are the expected values of $P_1, ..., P_m$.*

*Proof.* The regret is defined as follows:

$$\text{Regret} = \sum_{j:\Delta_j>0} \Delta_j \mathbb{E}[C_j(T)], \tag{6}$$

where $\Delta_i = Q^* - Q(s, i)$, $Q^* \doteq \max_i\{Q(s, i)\}$, and $C_i(T)$ denotes the selection number of machine $i$ over the first $T$ rounds. The key of the derivation lies in the decomposition of $\mathbb{E}[C_i(T)]$. Due to the initial evaluation conducted by UCB for each machine, $C_i(T)$ can be rearranged as:

$$C_i(T) = \sum_{t=1}^{T}\{I_t = i\} = 1 + \sum_{t=m+1}^{T} \{I_t = i\}, \tag{7}$$

where $\{\cdot\}$ is an indicator function. If the event $\{I_t = i\}$ occurs, meaning that the UCB policy selects machine $i$ in round $t$, then the indicator function outputs 1. Otherwise, it outputs 0.

Auer et al. (2002) derived the conclusion of the theorem through a step-wise approach using this equation. For a detailed derivation process, please refer to the original proof in (Auer et al., 2002). The subsequent derivations in this paper require only the few steps outlined above.  □

**Theorem 1.** *Suppose the player employs the node selection rule defined in (2). Then, in a scenario with $m$ available actions in state $s$, the bound of the expected cumulative regret is:*

$$\text{Regret} = \sum_{j:\Delta_j>0} \Delta_j \mathbb{E}[C_j(N(s))] \leq [8 \sum_{i:\Delta_i>0} (\frac{\ln N(s)}{\Delta_i})] + (1 + \frac{\pi^2}{3})(\sum_{i=1}^{m} \Delta_i), \tag{8}$$

*where $\Delta_i = Q^* - Q(s, i)$ for action index $i$ in $\{1, 2, ..., m\}$ and $Q^* \doteq \max_i\{Q(s, i)\}$, $N(s)$ is the visit count of $s$, and $C_i(N(s))$ is the selection number of action $i$.*

*Proof.* Since the definition of regret has not changed, we will continue our derivation starting from the decomposition of $C_i(T)$, where $T$ has the same meaning with $N(s)$. We will ultimately demonstrate that, due to the presence of the virtual node $y$, the player can achieve the same upper bound on regret as UCB. In the multi-armed bandit variant, action $i$ can be selected only if $y$ has been selected at least $i$ times. Thus, we define the selection number of $y$ at round $t$ as $A(t)$, and we have:

$$
\begin{aligned}
C_i(T) &= \sum_{t=1}^{T}\{I_t = i, A(t) \geq i\} = \sum_{t=i+1}^{T} \{I_t = i, A(t) \geq i\} \\
&= \sum_{t=i+1}^{T} \{I_t = i, A(t-1) \geq i\} \cup \{I_t = i, \xi(t-1, i)\} \\
&= \sum_{t=i+2}^{T} \{I_t = i, A(t-1) \geq i\} + \sum_{t=i+1}^{T} \{I_t = i, \xi(t-1, i)\},
\end{aligned} \tag{9}
$$

where $\xi(t, i)$ denotes the event in which player select $y$ for the $i$-th time at round $t$. The derivation in the first line follows from the fact that $A(1) = 0$ and $A(t) \leq t - 1$. Therefore, when $t \leq i$, the event $A(t) \geq i$ cannot occur. The derivation from the first line to the second line primarily relies on rewriting the original event $A(t) \geq i$ as the union of two mutually exclusive events, $A(t - 1) \geq i$ and $\xi(t - 1, i)$. By repeatedly applying the above decomposition operation to the first term of the formula, we have:

$$C_i(T) = \sum_{k=1}^{n-i} \sum_{t=i+k}^{n} \{I_t = i, \xi(t - k, i)\} = \sum_{t=i+1}^{n} \sum_{k=1}^{t-i} \{I_t = i, \xi(t - k, i)\}$$
$$= \sum_{t=i+1}^{n} \sum_{x=i}^{t-1} \{I_t = i, \xi(x, i)\} = \sum_{x=i}^{n-1} \sum_{t=x+1}^{n} \{I_t = i, \xi(x, i)\} \tag{10}$$

By setting $x = t - k$ and exchanging a series of inner and outer loops, we obtain a more interpretable formula. This formula first determines the round in which $y$ is selected for the $i$-th time by iterating the variable $x$ over the interval $[i, n - 1]$, followed by iterating the time $t$ to compute the selection number of action $i$. Clearly, the above expression is less than $n$, leading directly to a linear regret bound. However, to achieve the goal of deriving a sub-linear bound, we should use the following fact:

$$C_i(T) \leq \max_x \sum_{t=x+1}^{n} \{I_t = i, \xi(x, i)\} = \max_x \sum_{t=x+1, n} \{I_t = i\}$$
$$\leq \sum_{t=1}^{n} \{I_t = i\} \tag{11}$$

Thus, we can directly proceed to Equation (7) and its subsequent derivations, ultimately proving that in this case, the player can still achieve the same regret bound:

$$[8 \sum_{i:\Delta_i > 0} (\frac{\ln N(s)}{\Delta_i})] + (1 + \frac{\pi^2}{3})(\sum_{i=1}^{m} \Delta_i), \tag{12}$$

where $N(s)$ is equivalent to $T$. $\qquad\square$

In the calculation of the vUCB value in (3), the confidence bound for virtual nodes includes an additional coefficient $g(x)$ compared to that of actual nodes. Assuming the number of available actions is $m$, we expect the virtual node to be selected at most $m$ times, and the coefficient should decrease as the size of $\hat{\delta}(s)$ increases. Therefore, it should possess the following characteristics:

$$g(m) = 0 \text{ and } \frac{\partial g}{\partial x} <= 0 \tag{13}$$

However, since the number of available actions cannot be known in advance when planning in the LLM-based imagination space without environmental tools, the first property cannot be satisfied. Therefore, in practice, we use $g(x) = w_g(e^{-x^2} - \epsilon_g)$ as a substitute, whose value rapidly diminishes as the size of $\hat{\delta}(s)$ increases.

**Theorem 3.** *Let the expected TV-distance between two transition distributions be bounded at each timestep by $\epsilon_m$ and the policy divergence be bounded by $\epsilon_\pi$. Then the true returns and model returns of the policy are bounded as:*

$$\eta[\pi] \geq \hat{\eta}[\pi] - [\frac{2\gamma(\epsilon_m + 2\epsilon_\pi)}{(1 - \gamma)^2} + \frac{4r_{\max}\epsilon_\pi}{(1 - \gamma)}]. \tag{14}$$

In the above theorem, $\eta$ is the returns of the policy in the true environment, $\hat{\eta}$ is the returns of the policy under the LLM-based world model. $r_{\max}$ is the maximum value of rewards, $\epsilon_m = \max_{(s,a) \sim \mathcal{M}}[D_{\mathrm{TV}}(\mathbb{T}(\cdot|s, a))||(\mathbb{T}(\cdot|s, a; \theta))]$ is the generalization error of the LLM-based world model, $(s, a)$ is sampled from MLAQ's memory module $\mathcal{M}$, and $\mathbb{T}(\cdot|s, a; \theta))$ is the transition distribution of the LLM-based world model. $\epsilon_\pi \geq D_{\mathrm{TV}}(\pi||\pi_{\mathcal{M}})$ is the policy divergence between greedy policy $\pi$ derived from Q-learning (output argmax actions) and the data-collecting policy in the imagination space $\pi_{\mathcal{M}}$. Please refer to the proof of Theorem 3.1 in (Janner et al., 2019).

# D    EXPERIMENTAL ENVIRONMENTS

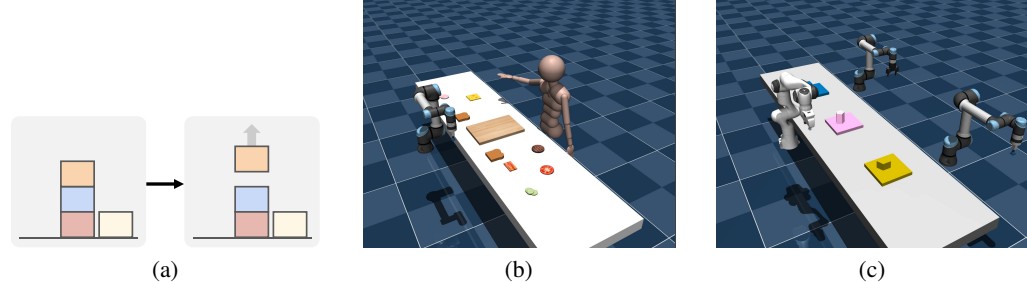

Figure 7: Experimental environments. (a) BlocksWorld. (b) Sandwich domain of RoCo-benchmark. (c) Sort domain of RoCo-benchmark.

As shown in Figure 7, we evaluate the methods in BlocksWorld (single-agent) and RoCo (multi-agent) benchmarks. Next, we will provide a detailed description of each task, as well as the state and action space.

## D.1    BLOCKSWORLD

BlocksWorld is a well-known single-agent benchmark for evaluating LLM agents (Valmeekam et al., 2022), where an agent should rearrange some blocks into multiple stacks in a particular order. In this work, we choose the tasks with four blocks to evaluate our methods. Hao et al. (2023) has grouped all tasks according to the optimal step, and we randomly choose up to 30 tasks from each group for evaluation. As shown in Figure 8 (a), the state space in the original BlocksWorld's codes has a lot of redundant information, which may help agents make better decisions, but may lead to errors when making predictions by the world model. Therefore, as shown in Figure 8 (b), we adjust the format of them in a more concise and clear manner without affecting the validity of states.

The **state** records what the agent's hand is holding: ('Empty' or 'Holding <block>') and what each block is on ('on <block>', 'on table', or 'in hand'). The **action** is an instruction that moves one block, which is one of STACK, UNSTACK, PUT, and PICK UP.

1. PICK UP <object>: Execute if hand is empty, and <object> is on table and no block is on it. Pick up <object> from table. After execution, the agent will be holding the <object>.

2. UNSTACK <object>: Execute if hand is empty, <object> is on another block and no block is on it. UNSTACK <object> from another block. After execution, the agent will be holding the <object>.

3. PUT DOWN <object>: Execute if hand is holding <object>. Put down <object> on table. After execution, the agent's hand will be empty.

4. STACK <object> ON <target>: Execute if hand is holding <object>, and no block is on <target>. Stack <object> on the top of <target>. After execution, the agent's hand will be empty.

## D.2    ROCO-BENCHMARK

RoCo-benchmark is built upon the Mujoco engine (Todorov et al., 2012) to construct several multi-arm scenarios. RoCo-benchmark provides 6 different tasks, including those containing two agents and those containing three agents, and the robotic arms in them have their own movable range and require full cooperation to complete the given task. We choose two of these tasks, Sort and Sandwich, to evaluate our method. In Roco-benchmark's codes, there is no fixed format for the state space, and the states are intermingled in the form of information within the agent's prompt. Therefore, we have constructed the state spaces for these two tasks in a concise manner, while keeping the action spaces consistent with the original codes.

**(a)**

I have that, the orange block is clear, the yellow block is clear, the hand is <state>, the blue block is <state>, the orange block is <state>, the red block is <state> and the yellow block is <state>.

**(b)**

[State]

[Block States]
blue block: <block state>
orange block: <block state>
red block: <block state>
yellow block: <block state>

[Hand State]
<hand state>

**(c)**

[State]

[Food States]
bread_slice1: <food state>
bread_slice2: <food state>
bacon: <food state>
cheese: <food state>
tomato: <food state>
cucumber: <food state>
ham: <food state>
beef_patty: <food state>

[Robot States]
1. Chad's gripper is <gripper state>
2. Dave's gripper is <gripper state>

**(d)**

[State]

[Cube States]
blue_square is on <location>
pink_polygon is on <location>
yellow_trapezoid is on <location>

Figure 8: State templates. (a) Template in original BlocksWorld codes. (b) Modified Template in BlocksWorld. (c) Template in Sandwich domain of RoCo-benchmark. (d) Template in Sort domain of RoCo-benchmark.

**Sandwich domain.** There are two robots, Chad and Dave, trying to make a sandwich following the given recipe order. Chad can only reach for food items on the right side of the table, while Dave can only reach for food items on the left side, with each food item starting in a fixed position. The robots need to place the food items on the cutting board in the middle of the table one by one, following the order given in a recipe. Different types of sandwiches contain different combinations of food items. Take the 'bacon' sandwich as an example, it requires 'bread_slice1', 'bacon', 'cheese', 'tomato', and 'bread_slice2'. In a bacon recipe, apart from the fixed positions of 'bread_slice1' and 'bread_slice2' as the two ends of the sandwich, the order of the other food items can vary across different recipes. Therefore, one type of sandwich may have multiple different recipes. We group all the recipes according to the optimal step to evaluate the agents. In the decision-making process, robots cannot execute PUT actions simultaneously within one timestep to avoid collisions.

As shown in Figure 8 (c), the **state** records the state of each food item ('on left/right side', 'atop <another food item> or <cutting_board>', or 'gripped by <robot>') and the state of each robot's gripper ('holding <food item>' or 'empty'). The **action** is an instruction for each agent to interact with the food items, which is one of PICK, PUT, and WAIT.

1. PICK <obj>: Execute if gripper is empty and the food item's state is not 'atop <obj>'. Pick the <obj> from the table. After execution, the robot's gripper will be holding <obj>.

2. PUT <obj1> <obj2>: Execute if gripper is holding <obj1>. Put <obj1> on the top of <obj2>. After execution, the robot's gripper will be empty.

3. WAIT: Do nothing.

The action checker in the Sandwich environment strictly limiting the placement of food items to follow the recipe, rather than allowing it to be placed in an incorrect order. Therefore, the size of agents' available action set in each state is relatively small. The agent can only choose to wait, pick up the next food item, or place the food item from the gripper according to the recipe, so that RoCo could achieve a high success rate in this task, but the optimal rate is still low.

**Sort domain.** There are three robots, Alice, Bob, and Chad, trying to sort three cubes onto their corresponding panels. There are a total of seven panels. Alice can access panels 1 to 3, Bob can access panels 3 to 5, and Chad can access panels 5 to 7. The three cubes are a blue square, a pink polygon, and a yellow trapezoid, with their respective target panels being panel2, panel4, and panel6. In the decision-making process, a panel cannot contain two or more cubes, and a cube cannot be grasped by two robots simultaneously.

As shown in Figure 8 (d), the **state** records the state of three cubes (on <panel>) and the action is an instruction for each agent to interact with the cubes, which is one of PICK & PLACE and WAIT.

1. PICK <object> PLACE <target>: Pick up <object> and place it onto <target>, where <object> is a cube and <target> is a panel.
2. WAIT: Do nothing.

Given the complexity of this domain involving more agents and expansive available action spaces, its difficulty surpasses that of the Sandwich domain. In the Sandwich domain, decision trajectories are linear and decision loops are non-existent, meaning that the multi-agent system cannot revert to its initial state after a series of decisions. However, in the Sort domain, the occurrence of loops adds to the challenge of reaching target states. MLAQ could address this challenge by leveraging RL-based optimization to make long-term optimal decisions.

## E    EXPERIMENTS ON CRAFTER

To validate MLAQ's performance in partially observable scenarios with high complexity, we chose Crafter (Hafner, 2022) (a 2D version of MineCraft) as our experimental environment.

In Crafter, players need to mine diamonds on a 64 * 64 map. To achieve this goal, there are 16 sub-goals to complete (collecting woods, stones, make wood pickaxe, etc.), of which 9 are essential for diamond mining, making it a challenging task with extremely long decision sequences. Additionally, this is an extremely difficult sparse reward problem, where reward signals are only provided when successfully achieving a sub-goal.

Players can only observe a 9 * 9 local map centered on their position, and we adopted SmartPlay's (Wu et al., 2024) natural language observation setting: for multiple items within the field of view, only the nearest one will be shown in the observation. For example, even if there are 3 trees in view, the observation will only show "tree is 3 steps to your north-east", which further intensifies the partial observability of Crafter.

To achieve optimal decision-making in this task (and other stochastic problems), mainly because the same state-action pair may lead to different next states, we made two minor modifications to MLAQ:

1. We remove the Prediction Checker, as it's difficult to provide effective verification in the cases with stochasticity.
2. We remove MLAQ's re-utilization of memory transitions during planning, because the same state-action pair may lead to multiple next states, and direct re-utilization would lose stochasticity. However, this can be adjusted as needed in practical, such as starting random re-utilization when the number of next states corresponding to the same state-action pair exceeds a threshold.

In table 7, we compare MLAQ with several existing LLM agents, including EnvGen (Zala et al., 2024) and DiVE (Sun et al., 2024), and the results of RL agents and human are from (Sun et al., 2024).

Table 7: The scores of experimental methods in Crafter domain.

| Method | Human | MLAQ-gt | MLAQ-script | MLAQ | DiVE | EnvGen | Dreamer-V3 | PPO | Random |
|--------|-------|---------|-------------|------|------|--------|------------|-----|--------|
| Scores | 50.5% | 46.2%   | 42.6%       | 39.9% | 35.9% | 32.2% | 14.5%      | 4.6% | 1.6%   |

We calculate The scores using the same methodology as existing works, which is determined by the ratio of unblocked achievements. The data of MLAQ are obtained using the same testing methods

as SmartPlay (Wu et al., 2024). In Crafter, the move action exhibits strong stochasticity, making it impossible for players to infer post-movement observations from local information. We conducted three experiments:

1. MLAQ-gt utilizes ground-truth environment simulation for move actions, providing the exact next state for given state-action pair, thus effectively eliminating movement stochasticity.

2. MLAQ-script generates post-movement observations through scripted randomization (e.g., after moving north, nine unknown areas appear at the northernmost position, with randomly assigned items like stone, sand).

3. MLAQ reconstructs the aforementioned script into prompts, allowing LLM to fully implement an LLM-based World Model.

The results align with expectations: MLAQ-gt achieves best performance using ground-truth environment, consistently crafting iron pickaxes but struggling to explore the location of diamonds within limited steps in 64 * 64 maps. MLAQ-script and MLAQ perform similarly but significantly below MLAQ-gt, indicating stochasticity substantially impacts MLAQ, though still outperforming other existing LLM agents and RL agents. It's worth noting that these experimental data are averaged from eight tests, serving to demonstrate to reviewers the potential of MLAQ in complex environments.

## F    Pseudo-code for the Overall Algorithm

We present detailed pseudo-code for our MLAQ framework in this section, and these algorithms are used to fully describe the overall process presented in Figure 1 and Figure 2. We have simplified the entire algorithm process into the flowchart in Figure 9 to facilitate a better understanding of how MLAQ makes optimal decisions for new a task.

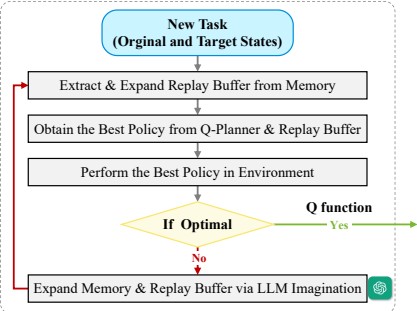

Figure 9: Flowchart of MLAQ agent to obtain the optimal decision sequence for a new task, with the input being the domain description and domain-specific memory and the output being the Q function.

### F.1    Overall algorithm for MLAQ agent

At first, Algorithm 1 shows how MLAQ agent obtains a optimal decision sequence in a given task. Its inputs are the environment dynamics and reward function with the domain description in natural language $\tau$ to provide interaction between the true environment and the agent (or multi-agent system), which is also depicted in the left side of Figure 1 (a). At the beginning of the optimization of MLAQ agent, we initialize an empty memory $\mathcal{M}$ and a Q function $Q(\cdot, \cdot)$ to set all Q values to -1. As described in the experiment section, we then scan all tasks of a domain to gradually expand the memory module through environmental and imaginary interactions.

For each single task, we have a unique pair of original state $s_0$ and target state $\bar{s}$, and initialize a replay buffer $\mathcal{D}$ for this task. Before making decisions in this task, MLAQ agent will try to find this task's best trajectory from the memory to expand the empty replay buffer, and a task-specific Q function $Q(\cdot, \cdot)$ will be obtained based on $\mathcal{D}$. If the agent fails in finding an available trajectory from the memory, meaning that the $Q(s_0, a)$ is no larger than the threshold $\tilde{Q}$ for any action $a$, we would skip the following environmental interaction phase to directly perform the imagination process until an available decision sequence is found. Otherwise, the agent interacts with the environment by making best decisions according to $Q(\cdot, \cdot)$ to evaluate the optimality $\mathbb{1}(s_0, \bar{s})$ of the resulting decision sequence. The indicator $\mathbb{1}(s_0, \bar{s})$ is set to True only if the environmental sequence is optimal.

In order to expand the memory and replay buffer to optimize the agent, we keep performing imaginary interactions until the optimal sequence is obtained for the indicator $\mathbb{1}(s_0, \bar{s})$ is True or the maximum trial number $K_c$ is reached. As more tasks are scanned, the size of memory $\mathcal{M}$ will gradually increase, and we will finally output it to assist the MLAQ agent in making optimal decisions under any tasks in this domain. While for the tasks have not been scanned before, MLAQ would also use the transitions stored in $\mathcal{M}$ to reduce LLM queries and obtain the optimal decision sequence much easier.

### F.2    LLM-based Imagination for MLAQ

Algorithm 2 input the original state $s_0$ and target state $\bar{s}$ with domain description $\tau$. The memory $\mathcal{M}$ contains transitions from other tasks and the temporary replay buffer $\mathcal{D}$ is specific to the current task. At the beginning, we should first add a virtual node (if does not exist) for the original state $s_0$ to start the imaginary interactions. Then, we perform the phases of node selection and transition imagination to expand the memory and replay buffer, and the overall process has already described in section 3. The outputs of the imaginary interaction are the expanded memory and replay buffer. It should be noted that after each step of transition imagination, we would perform an "Expand-Buffer" method to find an available trajectory with better optimality within $\mathcal{M}$ and add nodes along this trajectory to $\mathcal{D}$.

---

**Algorithm 1:** Obtaining optimal decision sequence for MLAQ agent

---

1 **Inputs:**
2     The environment, including dynamics $s' = \mathbb{T}(s, a)$ and reward function $r = \mathbb{R}(s, a, s')$;
3     Task description in natural language $\tau$;
4 **Initialize:**
5     memory $\mathcal{M}$ and Q function $Q(\cdot, \cdot) = -1$;
6 **for** task $= 1, 2, 3, ...$ **do**
7  | *# Each task has a unique original state $s_0$ and target state $\bar{s}$*
8  | Initialize a replay buffer $\mathcal{D}$ for this task;
9  | Set the optimal sequence indicator for this task $\mathbb{1}(s_0, \bar{s})$ to be False;
10 | **while** $\mathbb{1}(s_0, \bar{s})$ is not True **AND** trial $< K_c$ **do**
11 |  | *# Expand the replay buffer and get a task-specific Q function based on the memory*
12 |  | $\mathcal{D} = \texttt{Expand-Buffer}(s_0, \bar{s}; \mathcal{M}, \mathcal{D})$;
13 |  | $Q(\cdot, \cdot) = \texttt{Q-Update}(s_0, \bar{s}; \mathcal{D})$;
14 |  | **if** $\max_a Q(s_0, a) < \tilde{Q}$ **then**
15 |  |  | *# An available trajectory does not exist based on $\mathcal{D}$*
16 |  |  | Skip the environmental interaction phase and turn to the next trial of imagination;
17 |  | **end**
18 |  | *# Environmental Interaction Phase*
19 |  | **while** $s_t$ is not $\bar{s}$ **AND** $t < T$ **do**
20 |  |  | Get the optimal action $a_t = \arg\max_a Q(s_t, a)$;
21 |  |  | **if** $a_t$ is available for the environment **then**
22 |  |  |  | Perform action to the environment $s_{t+1} = \mathbb{T}(s_t, a_t)$ and $r_t = \mathbb{R}(s_t, a_t, s_{t+1})$;
23 |  |  | **else**
24 |  |  |  | Remove transition $(s_t, a_t, r(s_t, a_t), c(s_t, a_t))$ from $\mathcal{M}$ and $\mathcal{D}$;
25 |  |  |  | Get environmental feedback $\mathcal{F}$ and store $(s_t, a_t, \mathcal{F})$ into $\mathcal{M}$ and $\mathcal{D}$;
26 |  |  | **end**
27 |  |  | Store transition $(s_t, a_t, r_t, s_{t+1})$ into $\mathcal{M}$ and correct the wrong transition;
28 |  |  | $s_t \leftarrow s_{t+1}$;
29 |  | **end**
30 |  | **if** $s_t$ is $\bar{s}$ **then**
31 |  |  | $\mathbb{1}(s_0, \bar{s})$ is True **if** length of decision sequence is optimal **else** False;
32 |  | **else**
33 |  |  | $\mathbb{1}(s_0, \bar{s})$ is False;
34 |  | **end**
35 |  | **if** $\mathbb{1}(s_0, \bar{s})$ is not True **then**
36 |  |  | *# Perform a round of imaginary interaction to expand the replay buffer and memory*
37 |  |  | $\mathcal{M}, \mathcal{D} = \texttt{Imagination}(s_0, \bar{s}; \tau, \mathcal{M}, \mathcal{D})$;
38 |  | **end**
39 | **end**
40 **end**
41 **Outputs:**
42     memory $\mathcal{M}$;

---

---

**Algorithm 2:** `Imagination`$(s_0, \bar{s}; \tau, \mathcal{M}, \mathcal{D})$ guided by UCB values

---

1  **Inputs:**
2     Original state $s_0$ and target state $\bar{s}$ of this task with domain description in natural language $\tau$;
3     The memory $\mathcal{M}$ and replay buffer $\mathcal{D}$;
4  **Initialize:**
5     Add a virtual node as the first child node of the original state $s_0$, and $N(s_0) \leftarrow N(s_0) + 1$;
6  **while** $s_t$ is not $\bar{s}$ **AND** $t < T$ **do**
7     *# Node Selection Phase*
8     Get the optimal action $a_t = \arg\max_a \mathrm{vUCB}\big(s_t, a, c(s_t, a)\big)$;
9     **if** $c(s_t, a_t)$ is a virtual node **then**
10        *# Transition Imagination Phase*
11        Action available indicator $\hat{\mathbb{1}}(a_t) \leftarrow$ False;
12        *# Basic policy makes decisions and action checker verifies if the output is available*
13        **while** $\mathbb{1}(a_t)$ is not True **AND** trial $< K_a$ **do**
14           Get a non-virtual action through LLM-based basic policy $a_t \sim \pi(s_t; \tau, \mathcal{M})$;
15           **if** $(s_t, a_t)$ in memory $\mathcal{M}$ **then**
16              Retrieve transition data $\big(s_t, a_t, r(s_t, a_t), c(s_t, a_t)\big)$ from $\mathcal{M}$ to $\mathcal{D}$;
17              $\hat{\mathbb{1}}(a_t) = $ True;
18           **else**
19              Check the action $\hat{\mathbb{1}}(a_t) = \xi_a(s_t, a_t; \tau, \mathcal{M})$;
20           **end**
21        **end**
22        $\triangleright$   This trajectory is failed and terminated **if** $\mathbb{1}(a_t)$ is False;
23        **if** $\mathbb{1}(a_t)$ is True **then**
24           *# World model predicts and prediction checker verifies if the output is available*
25           Prediction available indicator $\hat{\mathbb{1}}(s_{t+1}) \leftarrow$ False;
26           **while** $\hat{\mathbb{1}}(s_{t+1})$ is not True **AND** trial $< K_s$ **do**
27              Predict the next state $s_{t+1} \sim \hat{\mathbb{T}}(s_t, a_t; \tau, \mathcal{M})$;
28              Check the prediction $\hat{\mathbb{1}}(s_{t+1}) = \xi_s(s_t, a_t, s_{t+1}; \tau, \mathcal{M})$;
29           **end**
30        **end**
31        $\triangleright$   This trajectory is failed and terminated **if** $\mathbb{1}(s_{t+1})$ is False;
32     **else**
33        $s_{t+1} = c(s_t, a_t)$;
34        **if** $s_{t+1}$ is not in $\mathcal{D}$ **then**
35           Add a virtual node as the first child node of $s_{t+1}$;
36        **end**
37     **end**
38     $s_t \leftarrow s_{t+1}$ and $N(s_t) \leftarrow N(s_t) + 1$;
39     **if** $s_t$ is $\bar{s}$ **then**
40        $r_t = +1$;
41     **else if** $s_t$ is a failure state **then**
42        $r_t = -1$;
43     **else**
44        $r_t = 0$;
45     **end**
46     *# Expand the replay buffer and memory*
47     Store transition $(s_t, a_t, r_t, s_{t+1})$ into the replay buffer and memory (if does not exist);
48     *# Expand the replay buffer from memory if possible*
49     $\mathcal{D} = $ `Expand-Buffer`$(s_0, \bar{s}; \mathcal{M}, \mathcal{D})$;
50  **end**
51  **Outputs:**
52     The expanded memory $\mathcal{M}$ and replay buffer $\mathcal{D}$;

---

### F.3 Expand the replay buffer from the memory

Algorithm 3 is used to expand the task-specific replay buffer based on the transitions in the memory. It firstly finds an available trajectory with better optimality from $\mathcal{M}$ and then store the transitions of this trajectory in the replay buffer $\mathcal{D}$. This will effectively improve the exploration efficiency of the MLAQ agent within the imagination space. For example, if the replay buffer is empty and there exists an available trajectory from $s_0$ to $\bar{s}$, then the nodes and edges in this trajectory will be added to $\mathcal{D}$. Therefore, the first trial in Algorithm 1 would not require any LLM queries to get an available policy for this task.

Furthermore, if $\mathcal{D}$ is not empty, this algorithm could also help the MLAQ agent to aggregate transitions from memory for enhancing optimal decision-making capability in complex tasks. For example, if the agent explores a new transition and the next state corresponding to this transition has a trajectory with better optimality towards the target state $\bar{s}$, the MLAQ agent could add the nodes in this subsequent trajectory to $\mathcal{D}$ without extra exploration within the imagination space.

---

**Algorithm 3:** `Expand-Buffer`$(s_0, \bar{s}; \mathcal{M}, \mathcal{D})$

---

1 **Inputs:**
2     Original state $s_0$ and target state $\bar{s}$ of this task with task description in natural language $\tau$;
3     The memory $\mathcal{M}$;
4 $Q(\cdot, \cdot) =$ `Q-Update`$(s_0, \bar{s}; \mathcal{M})$;
5 **if** $\max_a Q(s_0, a) > \tilde{Q}$ **then**
6     $s_t \leftarrow s_0$;
7     **while** $s_t$ is not $\bar{s}$ **do**
8         Get the optimal action $a_t = \arg\max_a Q(s_t, a)$;
9         Retrieve transition data $\big(s_t, a_t, r(s_t, a_t), c(s_t, a_t)\big)$ from $\mathcal{M}$ to $\mathcal{D}$ (if does not exist);
10         $s_t \leftarrow c(s_t, a_t)$;
11     **end**
12 **end**
13 **Outputs:**
14     The expanded replay buffer $\mathcal{D}$;

---

### F.4 Q FUNCTION UPDATE

Algorithm 4 is employed to get a Q function for the given task $s_0$ and $\bar{s}$ on the basis of $\mathcal{X}$, where $\mathcal{X}$ could be the memory $\mathcal{M}$ and the replay buffer $\mathcal{D}$. At first, we initialize the Q function to set all Q values to -1, and assign rewards for all transitions based the target state $\bar{s}$. Then, the algorithm would perform multiple loops of Q-learning process to get the approximated Q function $Q(\cdot, \cdot)$. This algorithm will be used by the Q-Planner in Algorithm 1 and the replay buffer expansion process in Algorithm 3.

---

**Algorithm 4:** `Q-Update`$(s_0, \bar{s}; \mathcal{X})$

---

1 **Inputs:**
2     The original state $s_0$ and target state $\bar{s}$ of the given task;
3     The memory module or replay buffer $\mathcal{X}$;
4 **Initialize:**
5     Initialize a Q function $Q(\cdot, \cdot) = -1$;
6     For all transition $(s, a, r, s')$ in $\mathcal{X}$, if $s'$ is $\bar{s}$, then $r = +1$; if $s'$ is a failure state, then $r = -1$; otherwise, $r = 0$;
7 **for** loop $k = 1, 2, 3, ...$ **do**
8     **for** $(s_t, a_t, r_t, s_{t+1}) \in \mathcal{X}$ **do**
9         Perform the following Q-learning update rule for this transition;
10

$$Q_{k+1}(s_t, a_t) = Q_k(s_t, a_t) + \alpha \left( r_t + \gamma \max_{a'} Q_k(s_{t+1}, a') - Q_k(s_t, a_t) \right)$$

11     **end**
12 **end**
13 **Outputs:**
14     The Q function $Q(\cdot, \cdot)$;

---

## G  DETAILED PROMPT TEMPLATE

This section presents the detailed prompt templates for the basic policy, world model, action checker, and prediction checker. The prompt template has multiple parts to provide different types of information for the modules. The prompts in the purple cube are the component-specific prompts to guide the agent in making decisions, predicting next states, checking actions or predictions. In the prompt, we guide the corresponding modules step-by-step to enhance the accuracy of their outputs. The prompts in the green cube contain some task-specific information, and prompts in the blue cube contain some self-provided information generated from self-examinations and vUCB-based planning. The prompts in grey cube is the 'user prompt', which contains the current information for the modules.

### G.1  BASIC POLICY PROMPT

The prompt shown in Figure 10 requires the current state and target state in natural language to make decisions. In the **[Detailed Instruction for Policy]** part, we decompose the decision-making process into five parts: [Action Planning], [Action Conclusion], [Action Constraints Check], [Forbidden Action Check], and [Action Output], gradually guiding the agent to make decisions that satisfy the constraints. As for the basic policy of multi-agent systems, we directly follow the prompt design of RoCo, and the prompts can be found in (Mandi et al., 2023). The policy mistakes are the environmental feedbacks of previous wrong actions stored in the memory $\mathcal{M}$. The policy feedbacks are the temporary feedbacks provided by the self-examination only under the current state. The forbidden actions are the non-virtual actions of the current state node's child nodes.

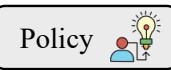

You are a planner. You have received a [State] from the external environment. Now, you need to output an action to reach the target [State] from current [State]. Follow the [Detailed Instruction for Policy] strictly.

**[Detailed Instruction for Policy]** ...
You should output your thinking process step-by-step in the following order: [Action Planning], [Action Conclusion], [Action Constraints Check], [Forbidden Action Check], and [Action Output] ...

The following lists the [Task Description], [State Space Definition], and [Action Space Definition].
**[Task Description]** ...
**[State Space Definition]** ...
**[Action Space Definition]** ...

**[Policy Mistakes]** (long-term)
In the current state, you have chosen a wrong [Action], and the environment provides the [Feedback]:
**[Wrong actions and feedbacks]** ...

**[Policy Feedbacks]** (short-term)
The following actions are the wrong actions you have made. These feedbacks are from the previous imagination, and may not be correct. However, you still need to pay attention to them and avoid making the same or similar mistakes.
**[Wrong actions and feedbacks]** ...

**[Forbidden Actions]**
You should try your best to output an action that is not in the list below ...
**[Action List]**

Now you should output your action according to the following current state and target state.
**{state}**
**{target_state}**

Figure 10: A simplified template for the basic policy prompt.

## G.2 ACTION CHECKER PROMPT

The prompt shown in Figure 11 requires the current state and the action should be checked. In the **[Checker Instruction]** part, we decompose the checking process into three parts: [Action Constraints Check], [Conclusion Justification], and [Checker Conclusion]. We provide the previous mistakes and the corresponding feedbacks stored in the memory under this current state for the action checker to avoid the same mistakes.

Figure 11: A simplified template for the action checker prompt.

## G.3 WORLD MODEL PROMPT

The prompt shown in Figure 12 requires the current state and action to predict the next state. In the **[Predicting Instruction]** part, we decompose the predicting process into three parts: [Interaction Item Pool], [Action Forward Rule] and [Prediction Conclusion]. The prompt also contains the feedbacks from the long-term and replay buffer. In fact, LLM can better handle the task of predicting with step-by-step reasoning, and we also find the prediction accuracy of the world model to be nearly 100% in the experimental section.

## G.4 PREDICTION CHECKER PROMPT

The prompt shown in Figure 13 requires the current state, executed action, and the prediction should be checked. In the **[Checker Instruction]** part, we decompose the checking process into three parts: [Prediction Format Check], [Prediction Rule Check], and [Checker Conclusion]. We also provide the previous mistakes and the corresponding feedbacks stored in the memory under this state-action pair for the prediction checker to avoid the same mistakes.

World Model 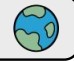

You are a state predictor. You will be given a [State] and an [Action] as input. should predict the [State] after the [Action] is applied to the given [State]. You should follow the [Predicting Instruction] to predict the [State].

**[Predicting Instruction]**
Please output your thinking process step-by-step by following theses steps: [Interaction Item Pool], [Action Forward Rule] and [Prediction Conclusion] ...

The following lists the [State Space Definition].
**[State Space Definition]** ...
**[State Template]** ...
**[Detailed Constraints for State]** ...

**[Prediction Mistakes]** (long-term)
In the current state and action, you have outputted a wrong prediction as follows:
**[Wrong predcitions and feedbacks]** ...

**[Prediction Feedbacks]** (short-term)
I have send your prediction to a GPT-based checker to check whether your prediction is correct. The checker said that your prediction is incorrect ...
**[Feedbacks]** ...

Now, you should predict the next [State] after the following [Action] is applied to the given [State]. You have to follow the instructions strictly rather than your own thoughts.
**{state}**
**{action}**

Figure 12: A simplified template for the world model prompt.

Prediction Checker 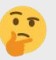

You are a prediction checker. You will receive a [State], an [Action], and the subsequent [Prediction]. You should check if the [Prediction] is valid based on the given [State] and [Action]. You should follow the [Checker Instruction] to check the [Action].

**[Checker Instruction]** ...
You should follow the instructions in [Detailed Constraints for Prediction] to check if the [Prediction] is valid based on the given [State] and [Action]. Take the mistakes in the [Prediction Mistakes] into consideration.

The following lists the [Detailed Constraints for Prediction].
**[Detailed Constraints for Prediction]**

**[Prediction Mistakes]** (long-term)
In the current state, you mistakenly believed that a wrong prediction was the correct one:
**[Wrong predictions and feedbacks]** ...

Now you should check the following [Prediction] based on the given [State] and [Action]. You have to check the instructions step-by-step strictly rather than your own thoughts.
**{state}**
**{action}**
**{predicion}**

Figure 13: A simplified template for the prediction checker prompt.

# H    DETAILED EXPERIMENT RESULTS

## H.1    BASIC POLICY ANALYSIS FOR MULTI-AGENT TASKS

As mentioned in the experiment section, our MLAQ agent framework is compatible with both single-agent and multi-agent scenarios. In multi-agent scenarios, it provides two basic policy decision paradigms. The first one treats the entire multi-agent system as a whole for centralized decision-making, where it takes the global state as inputs and outputs joint actions for the system. The second one adopts RoCo's dialog mechanism (Mandi et al., 2023), where the agents conduct multiple rounds of dialogue with each other and determine the joint actions when reaching consensus: $a_{i,t} \sim \pi_i(\cdot|o_{i,t}, \rho_{i,t})$, where $\rho_{i,t} = [d_{i,1}, ..., d_{i-1,j}]$ is the chat history between agents. $d_{i,j}$ represents the message output by agent $i$ at the dialog round $j$.

We evaluate the agents using these two basic policies in the Sort and Sandwich domains. As shown in Table 8, the results indicate that, with similar token consumption, the performance of centralized decision-making significantly decreases, with env replans increasing nearly tenfold in the Sandwich domain. The complexity of centralized decision-making significantly surpasses that of decentralized decision-making, thereby diminishing the effectiveness of basic policies in obtaining available or optimal actions.

Table 8: Performance comparison of using dialog and central paradigms in multi-agent domains.

| Task | Methods | Success Rate | Optimal Rate | Token | Env Replans | Optimal Gap |
|------|---------|--------------|--------------|-------|-------------|-------------|
| Sort | dialog | **0.98** | **0.75** | **175560** | 0.06 | **0.26** |
|  | central | 0.88 | 0.59 | 199753 | **0.04** | 0.57 |
| Sandwich | dialog | **0.95** | **0.45** | 208024 | **0.07** | **0.61** |
|  | central | 0.81 | 0.24 | **131244** | 0.62 | 3.24 |

## H.2    MAIN RESULTS IN THE SANDWICH DOMAIN

Due to space limitations in the main text, we provide the experimental results in the Sandwich domain in this section for reference. As shown in Table 9, although MLAQ is superior to RoCo in success rate and outperforms RoCo in long-horizon tasks (10-step), the token consumption of MLAQ is higher than that of RoCo. This is mainly due to two reasons. Firstly, the similarity between state transitions plays an essential role in token consumption, with tasks exhibiting higher similarity being better able to leverage stored transition data in memory to reduce token consumption. The Sandwich domain, with its relatively fixed trajectories compared to Sort, exhibits lower similarity across different tasks. This is evidenced by the average memory re-utilization ratio between the two domains, notably lower in Sort. This characteristic of the Sandwich domain not only contributes to RoCo achieving a higher optimal rate but also diminishes MLAQ's token saving capability. Secondly, MLAQ itself involves world model prediction and self-examination, leading to an average token consumption per time step exceeding RoCo's.

Table 9: Comparison of multiple metrics between MLAQ and RoCo in Sandwich domain.

| Metrics | Methods | 6-step | 8-step | 10-step | Average |
|---|---|---|---|---|---|
| Success Rate | RoCo | 1.00 | 0.64 | 0.92 | 0.75 |
| | MLAQ | **1.00** | **0.92** | **1.00** | **0.95** |
| Env Replans (n-shot) | RoCo | 2.50 | 3.87 | 4.33 | 3.86 |
| | MLAQ | **0.00** | **0.11** | **0.00** | **0.07** |
| Optimal Rate | RoCo | **0.75** | **0.36** | 0.25 | 0.37 |
| | MLAQ$^-$ | 0.00 | 0.08 | **0.83** | 0.29 |
| | MLAQ | 0.25 | 0.31 | **0.83** | **0.45** |
| Average Token | RoCo | **20740** | **35072** | **65027** | **42266** |
| | MLAQ$^-$ | 74881 | 63555 | 77949 | 68746 |
| | MLAQ | 184609 | 220521 | 188752 | 208024 |
| Optimal Gap | RoCo | **0.50** | **1.13** | 1.83 | 1.27 |
| | MLAQ$^-$ | 1.25 | 1.88 | **0.17** | 1.33 |
| | MLAQ | 0.75 | 1.35 | **0.17** | **0.96** |
| Memory Re-Util. Ratio | | 0.31 | 0.63 | 0.57 | 0.58 |

## I HYPER-PARAMETERS

Table 10 shows the hyper-parameters presented in the MLAQ training process. Except for the environmental horizon $T$, all other hyper-parameters remain constant across all experiments, where $T$ is set to 20 for the BlocksWorld experiment, 16 for the Sandwich experiment, and 8 for the Sort experiment.

In this paper, all experiments are conducted using the GPT API interface, without involving CPU or GPU usage. The total cost of the API resources used in this paper does not exceed 1500 US dollars, including preliminary tests, comparative experiments, and ablation experiments.

Table 10: Hyper-parameters presented in the MLAQ training process

| Hyper Parameter | Value |
|---|---|
| LLM source | gpt-4-0125-preview |
| Learning rate $\alpha$ | 1.0 |
| Discount $\gamma$ | 0.995 |
| UCB weight $w_g$ | 2 |
| vUCB weight $w_g$ | 4 |
| vUCB threshold $\epsilon_g$ | 0 |
| Maximum trial number for imagination $K_c$ | 2 |
| Maximum trial number for prediction $K_s$ | 2 |
| Maximum trial number for policy $K_a$ | 2 |
| Q threshold $\tilde{Q}$ | 0.5 |
| Q update loops | 20 |
| Environmental horizon $T$ | 20 / 8 / 16 |

## J PROMPT EXAMPLE

### J.1 POLICY PROMPT AND RESPONSE OF THE STATE-BASED DECISION-MAKING

In the LLM agent constructed by our method MLAQ, both decision-making in single-agent scenarios and centralized decision-making in multi-agent scenarios rely on the global state. We use the prompt template in Figure 10 to query LLM and output agent actions or joint actions of the multi-agent system. In the following, we provide an example of a policy prompt in the Sort domain, which includes all the elements in the template shown.

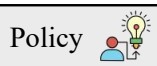

Policy

**[Detailed Instruction for Policy]**
You should output your thinking process step-by-step in the following order. You have to follow these steps one by one to plan the [Action]: [Action Planning], [Action Conclusion], [Action Constraints Check], [Forbidden Action Check], and [Action Output].
The most important thing: Follow the instructions step-by-step and ensure each step is completed precisely. Repeat the instructions and fill in the blanks '[]' without introducing any modifications or additional content.
 + **[Action Planning]**: Plan the [Action] of the multi-agent system step by step and list the thinking process.
 + **[Action Conclusion]**: Conclude the [Action] in the format of [Detailed Constraints for Action].
 + **[Action Constraints Check]**: Follow the steps in [Detailed Constraints for Action] to check the [Action] step by step.
 + **[Action Revise]**: If there is no incorrect checking result in the [Action Constraints Check], you can pass to the [Forbidden Action Check]. Otherwise, you should revise the action. Take the feedback from [Action Constraints Check] into consideration, and follow these steps to revise the action: [New Action Planning] and [ NewAction Constraints Check].
 + **[Forbidden & Mistake Action Check]**: I will provide some forbidden joint actions, and the robots cannot choose the forbidden and mistake actions. Note that what is forbidden are the joint actions of the multi-agent system, rather than the single agent actions that appear in these joint actions.
   - list the chosen action.
   - list all actions in the [Forbidden Actions] and [Policy Mistakes].
   - check if the [Action] is in the [Forbidden Actions] or [Policy Mistakes] one by one.
   If the [Action] is not in the [Forbidden Actions] and [Policy Mistakes]:
    - Pass to the [Action Output] part.
   Else:
    - Follow these instructions to plan a new [Actions] step by step and list the thinking process:
    - Go back to a new [Action Planning] and [Action Constraints Check] process again. You should note that what is forbidden are the joint actions rather than the single agent actions that appear in these joint actions. Therefore, the individual actions of each agent in the [Forbidden Actions] can still be chosen.
    - [Forbidden Action Check]: check if the [Action] satisfies the [Detailed Constraints for Action].
    - Re-plan the [Action] until a new [Action] is obtained or there is no available action anymore. You can try 2-nd, 3-rd, 4-th ... Plan to get the final [Action].
   If there is no available action anymore, output [ERROR] to denote that there is no available action anymore.
 + **[Action Output]**: output the final action in the format of [Detailed Constraints for Action].

The following lists the [Task Description], [State Space Definition], and [Action Space Definition].
**[Task Description]**
Task: Cooperative Sorting in a Multi-Agent System

Agents: Alice, Bob and Chad
Alice - Can only PICK and PLACE cube items on panel1, panel2 and panel3.
Bob - Can only PICK and PLACE cube items on panel3, panel4 and panel5.
Chad - Can only PICK and PLACE cube items on panel5, panel6 and panel7.

Objective: Collaboratively place the cubes on the panels as follows: place blue_square on panel2, place pink_polygon on panel4, place yellow_trapezoid on panel6. The cube items will be placed randomly on all panels of the table at the beginning of an episode.

**[State Space Definition]**
Define the state of the multi-agent system, which is composed of one category: [Cube States].
1. [Cube States]: Describe the status of the following three cube items: blue_square, pink_polygon, yellow_trapezoid.
2. The cube items must be listed in this order: blue_square, pink_polygon, yellow_trapezoid.

**[State Template]**
[State]
[Cube States]
blue_square is on <location>
pink_polygon is on <location>
yellow_trapezoid is on <location>

**[Detailed Constraints for State]**
You should specify the [State] according to the following constrains. A state is reasonable if it does not break the constraints one-by-one and step-by-step.
1. There must be three lines of [Cube States].
2. Check if their locations are one of: on panel1 - 7.
3. It is not allowed to have two cubes on the same panel. Check if they satisfy constraints.
3. The panel name should not contain " " in the middle. For example, use "panel2" instead of "panel 2".

**[Action Space Definition]**
Detail the action space for each robot with the stipulated actions: PICK PLACE, WAIT.
- PICK <object> PLACE <target>: Execute if the gripper is empty. Pick up <object> and place it onto <target>, where <object> is a cube and <target> is a panel
- WAIT: Do nothing.

**[Action Template]**
EXECUTE
NAME Alice ACTION <action>
NAME Bob ACTION <action>
NAME Chad ACTION <action>

**[Detailed Constraints for Action]**
**[Robot Reach Range]** Before showing the constraints, you should note that each robot has its own reach range, and it cannot PICK or PLACE items outside its reach range.
  + Alice: panel1, panel2 and panel3
  + Bob: panel3, panel4, panel5
  + Chad: panel5, panel6, panel7
**[Detailed Constraints]** Then, you should follow the constraints one-by-one and step-by-step to check if the action is correct: [Basic Constraints] and [PICK & PLACE Constraints]. You have to follow these constraints strictly and do not have your own understanding of the constraints.
The most important thing: Follow the instructions step-by-step and ensure each step is completed precisely. Repeat the instructions and fill in the blanks '[]' without introducing any modifications or additional content.
**[Basic Constraints]** Output "[Basic Check]" and do as follows:
  + There must be three lines of actions.
  + The agents cannot all perform WAIT.
  + Agents are not allowed to PICK <object> other than blue_square, pink_polygon, yellow_trapezoid.
  + Agents are not allowed to PICK the same <object> at the same time.
  + Agents are not allowed to PLACE objects on the same panel.
**[PICK & PLACE Constraints]** Now, output "[PICK & PLACE Check]" and do as follows:
  + For simplicity, we use [agent] to denote the agent's name: [Alice, Bob, Chad].
   - If action is PICK PLACE, follow the instructions step by step: [PICK Object Check], [PLACE Target Check].
     a. [PICK Object Check] Agents are not allowed to PICK objects out of its reach range.
     b. [PLACE Target Check] Agents are not allowed to PLACE objects out of its reach range.
     c. [PLACE Target Check] Agents are not allowed to PLACE objects on the panel that already has an object except it is PICKed by one of the agents at the current timestep.
  + For example: List Alice's action ... List Bob's action ... List Chad's action ...```

**[Policy Mistakes]**
No policy mistakes.

**[Temporary Mistakes]**
You have made 2 mistakes. The following actions are the wrong actions you have made.
These feedbacks are from the previous imagination, and may not be correct. However, you still need to pay attention to them and avoid making the same or similar mistakes.
**The wrong action 1 is as follows:**
EXECUTE
NAME Alice ACTION WAIT
NAME Bob ACTION PICK yellow_trapezoid PLACE panel6
NAME Chad ACTION WAIT
**The feedback is as follows:**
[ERROR] The action is invalid because it violates the reach range constraint for Bob, who cannot PLACE the yellow_trapezoid on panel6 as it is outside of his range.Now, output your action below. You should follow the [Detailed Instruction for Policy] strictly and think step-by-step to plan the action.

**The wrong action 2 is as follows:**
EXECUTE
NAME Alice ACTION WAIT
NAME Bob ACTION PICK yellow_trapezoid PLACE panel5
NAME Chad ACTION WAIT
**The feedback is as follows:**
[ERROR] Bob cannot PICK the yellow_trapezoid from its current location (panel1) because it is outside his reach range (panel3, panel4, panel5). This violates the PICK Range Check constraint.

**[Forbidden Actions]**
The following lists the forbidden actions:
**[Action 1]**
EXECUTE
NAME Alice ACTION PICK yellow_trapezoid PLACE panel3
NAME Bob ACTION WAIT
NAME Chad ACTION WAIT

Now you should output your action according to the following current state and target state.
**[State]**
[Cube States]
blue_square is on panel2
pink_polygon is on panel4
yellow_trapezoid is on panel1
**[Target State]**
[Cube States]
blue_square is on panel2
pink_polygon is on panel4
yellow_trapezoid is on panel6

The response of the basic policy is shown below. Although it already outputs a forbidden action list in the response, it finally outputs an action in this list, which indicates that the response of LLM is actually imperfect. Therefore, our MLAQ agent try to optimize through RL methods.

**[Action Planning]**
 - For Alice, since all her cubes are in the correct position, she should WAIT.
 - For Bob, since all his cubes are in the correct position or being taken care of by Chad, he should WAIT.
 - For Chad, since yellow_trapezoid needs to be moved from panel5 to panel6, and it's within his reach, he should PICK yellow_trapezoid PLACE panel6.

**[Action Conclusion]**
EXECUTE
NAME Alice ACTION WAIT
NAME Bob ACTION WAIT
NAME Chad ACTION PICK yellow_trapezoid PLACE panel6

**[Action Constraints Check]**
**[Basic Check]**
+ There must be three lines of actions: [correct].
+ The agents cannot all perform WAIT: [correct].
+ Agents are not allowed to PICK <object> other than blue_square, pink_polygon, yellow_trapezoid: [correct].
+ Agents are not allowed to PICK the same <object> at the same time: [no agent has the same <object>, correct].
+ Agents are not allowed to PLACE objects on the same panel. [no agent has the same <target>, correct].

**[PICK & PLACE Check]**
+ For Chad:
  - Chad's action: [PICK yellow_trapezoid PLACE panel6].
    a. [PICK Range Check] Agents are not allowed to PICK objects out of its reach range (Chad's: [panel5, panel6, panel7]): [no, action is incorrect].

**[Action Revise]**
Since the [PICK Range Check] for Chad's action was incorrect, the action needs to be revised. Chad cannot PICK the yellow_trapezoid from panel1 because it's out of his reach range. The correct action should involve Alice, who is in the range to PICK the yellow_trapezoid from panel1.

**[New Action Planning]**

Given the mistake identified during the [Action Constraints Check], we need Alice to move the yellow_trapezoid closer to Chad's range for him to eventually place it on panel6.

- Alice can PICK the yellow_trapezoid from panel1 and PLACE it on panel3, which is within her range and makes it accessible for Bob or Chad in subsequent actions.
- Bob and Chad will WAIT in this round to avoid any conflict.

**[New Action Conclusion]**

EXECUTE
NAME Alice ACTION PICK yellow_trapezoid PLACE panel3
NAME Bob ACTION WAIT
NAME Chad ACTION WAIT

**[New Action Constraints Check]**

**[Basic Check]**

+ There must be three lines of actions: [correct].
+ The agents cannot all perform WAIT: [correct].
+ Agents are not allowed to PICK <object> other than blue_square, pink_polygon, yellow_trapezoid: [correct].
+ Agents are not allowed to PICK the same <object> at the same time: [no agent has the same <object>, correct].
+ Agents are not allowed to PLACE objects on the same panel. [no agent has the same <target>, correct].

**[PICK & PLACE Check]**

+ For Alice's action: PICK yellow_trapezoid PLACE panel3.
   a. [PICK Range Check] Agents are not allowed to PICK objects out of its reach range (Alice's: [panel1, panel2, panel3]): [correct].
   b. [PLACE Range Check] Agents are not allowed to PLACE objects out of its reach range: [correct].
   c. [PLACE Target Check] Agents are not allowed to PLACE objects on the panel that already has an object except it is PICKed by one of the agents at the current timestep: [The panel3 is empty, correct].

**[Forbidden & Mistake Action Check]**

- list the chosen action: EXECUTE NAME Alice ACTION PICK yellow_trapezoid PLACE panel3 NAME Bob ACTION WAIT NAME Chad ACTION WAIT
- repeat all actions and feedbacks in the [Forbidden Actions], [Policy Mistakes], and [Temporary Mistakes]:
   a. [EXECUTE NAME Alice ACTION PICK yellow_trapezoid PLACE panel3 NAME Bob ACTION WAIT NAME Chad ACTION WAIT, forbidden action],
   b. [EXECUTE NAME Alice ACTION WAIT NAME Bob ACTION WAIT NAME Chad ACTION WAIT, [ERROR] The action is invalid because all agents cannot perform WAIT simultaneously.],
   c. [EXECUTE NAME Alice ACTION WAIT NAME Bob ACTION PICK yellow_trapezoid PLACE panel5 NAME Chad ACTION WAIT, [ERROR] Bob cannot PICK the yellow_trapezoid from its current location (panel1) because it is outside his reach range (panel3, panel4, panel5). This violates the PICK Range Check constraint.]]
- check if the [Action] is in the [Temporary Mistakes] one by one. If so, you should first judge whether the [Action] is correct or not. If the [Action] is correct, you can pass to the [Action Output] part. Otherwise, you should revise the [Action] step by step.
   + The chosen action is not listed in the [Temporary Mistakes].
- check if the [Action] is in the [Forbidden Actions] one by one.
   + The chosen action is not listed in the [Forbidden Actions].

**[Action Output]**

EXECUTE
NAME Alice ACTION PICK yellow_trapezoid PLACE panel3
NAME Bob ACTION WAIT
NAME Chad ACTION WAIT

The prompt and response in other tasks are also similar under the setting of state-based decision making.

## J.2 POLICY PROMPT AND RESPONSE OF THE OBSERVATION-BASED DECISION-MAKING

In the observation-based decision-making of LLM multi-agent systems, we fully utilize the RoCo's Policy Prompt to perform dialogues between agents, and only add an additional part to denote the forbidden actions. Here is an example in the Sort domain. We take the Alice's prompt as an example, the prompts of the other agents are also similar.

---

Alice's Policy 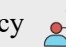

**[Action Space Definition]**
- PICK <object> PLACE <target>: pick up <object> and place it onto <target>, where <object> is a cube and <target> is a panel
- WAIT: Do nothing.

**[Action Template]**
EXECUTE
NAME Alice ACTION <action>
NAME Bob ACTION <action>
NAME Chad ACTION <action>

**[Action Output Instructions]**
1. Commence the output with 'EXECUTE
'. Follow with one distinct action per robot on separate lines.
2. Alice's action should be listed first, followed by Bob's action, and then Chad's action.
3. Agents' reach ranges are as follows:
  + Alice: zone1=[panel1, panel2, panel3]
  + Bob: zone2=[panel3, panel4, panel5]
  + Chad: zone3=[panel5, panel6, panel7]

You are robot Alice in front of panel2. You are collaborating with Bob, Chad to sort cubes into their target panels
Bob is tasked to place pink_polygon on panel4, and can reach panel3, panel4, panel5
Chad is tasked to place yellow_trapezoid on panel6, and can reach panel5, panel6, panel7
The task is NOT done until all three cubes are sorted correctly
There are 7 panels, ordered left to right: panel1,...,panel7. They form a straight assembly line, panel1 is closed to panel2 and farthest from panel7
Your goal is to place blue_square on panel2, but you can only reach panel1, panel2, panel3: this means you can only pick cubes from these panels, and can only place cubes on these panels
At current round:
blue_square is on panel5
pink_polygon is on panel4
yellow_trapezoid is on panel6
Never forget you are Alice!
1. You can only pick these reachable cubes:
2. You can only place them on these reachable panels: panel1, panel2, panel3
Think step-by-step about the task and others' response. Carefully check and correct them if they made a mistake
Improve your plans if given [Environment Feedback]

When you respond, tell others about your goal and all constraints. Respond very concisely but informatively, and do not repeat what others have said
Discuss with others to come up with the best plan, e.g. if your cube is out of your reach, ask others for help, and you can do the same for them
Propose exactly one action for yourself at the **current** round, select from [Action Options]
End your response by either: 1) output PROCEED, if the plans require further discussion, or 2) If everyone has made proposals and got approved, output EXECUTE
 and the final plan, must strictly follow [Action Output Instruction]!

You are Alice, before you make decisions, you should check if the [Action] is violating the [Forbidden Actions].
Your response is:

Since my target is to PUT ... on ..., and other robots ...

---

Here we provide a dialogue result from the above scenario, where three agents reach the consensus on each other's decisions through two rounds of dialogue. Finally, Bob summarizes and outputs the final joint action.

Dialogues 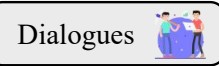

**Alice - Round 1**

Since my target is to place the blue_square on panel2, and other robots have their respective targets with Bob needing to place the pink_polygon on panel4 and Chad needing to place the yellow_trapezoid on panel6, we need to coordinate our actions. The blue_square is currently out of my reach on panel5, so I will need Bob's help to move it into my reach. Bob can pick up the blue_square from panel5 and place it on panel3, which is within both of our reach ranges. This action does not violate any forbidden actions as it involves moving a cube to an intersection panel to facilitate its transfer between zones. After Bob moves the blue_square to panel3, I can then pick it up and place it on its target panel, panel2, in a subsequent round.

Bob and Chad can proceed with their tasks as planned, with Bob moving the pink_polygon to panel4 as intended, and Chad placing the yellow_trapezoid on panel6. However, for this round, Bob's priority should be to assist me by moving the blue_square closer.

Let's discuss this plan and make sure it aligns with everyone's understanding and capabilities. If we all agree, we can proceed with the execution.

**PROCEED**

**Bob - Round 1**

Since my target is to place the pink_polygon on panel4, and other robots have their respective targets with Alice needing to place the blue_square on panel2 and Chad needing to place the yellow_trapezoid on panel6, we need to coordinate our actions carefully. The blue_square is currently on panel5, which is within my reach, and Alice has requested my help to move it into her reach. I can pick up the blue_square from panel5 and place it on panel3, which is within both Alice's and my reach ranges. This action does not violate any forbidden actions as it involves moving a cube to an intersection panel to facilitate its transfer between zones. After I move the blue_square to panel3, Alice can then pick it up and place it on its target panel, panel2, in a subsequent round.

My pink_polygon is already on panel4, which means it is correctly placed, and I do not need to move it this round. Chad can proceed with his task to place the yellow_trapezoid on panel6, which is already in the correct position as well.

Given Alice's plan and the current situation, my proposal for this round is to assist Alice by moving the blue_square closer to her. This will allow us to progress towards completing our overall task more efficiently.

Let's confirm if this plan works for everyone, including the movement of the blue_square to panel3 by me, and if there are no objections or further suggestions, we can proceed with the execution.

**PROCEED**

**Chad - Round 1**

Since my target is to place the yellow_trapezoid on panel6, and other robots have their respective targets with Alice needing to place the blue_square on panel2 and Bob needing to place the pink_polygon on panel4, we need to coordinate our actions carefully. The yellow_trapezoid is already correctly placed on panel6, which means my task for this round does not require any action on my part regarding the yellow_trapezoid.

Alice has proposed a plan for Bob to move the blue_square from panel5 to panel3, which is a feasible action and aligns with the goal of getting the blue_square closer to its target panel, panel2. This action does not violate any forbidden actions as it involves moving a cube to an intersection panel to facilitate its transfer between zones. Since my cube is already in the correct position, my focus this round should be on ensuring the plan aligns with the overall goal and does not interfere with my ability to maintain the yellow_trapezoid on panel6.

Bob has agreed to Alice's plan and has proposed to move the blue_square to panel3 to assist Alice. This action aligns with the collective goal of sorting all cubes into their target panels efficiently.

Given the current situation and the proposed actions by Alice and Bob, my action for this round will be to WAIT, as my cube is already in the correct position, and there are no immediate actions required from me to assist with the movement of the blue_square. My priority is to ensure the yellow_trapezoid remains on panel6 and to support Alice and Bob in their tasks as needed without violating any forbidden actions.

Let's confirm if this plan works for everyone, including my decision to WAIT this round, and if there are no objections or further suggestions, we can proceed with the execution.

**PROCEED**

The prompt and response in Sandwich domain are also similar under the setting of observation-based decision making.

### Alice - Round 2

Since my target is to place the blue_square on panel2, and other robots have their respective targets with Bob needing to place the pink_polygon on panel4 and Chad needing to place the yellow_trapezoid on panel6, we need to coordinate our actions carefully. The blue_square is currently on panel5, which is out of my reach. I need Bob's assistance to move it into my reach. However, upon reviewing the constraints and forbidden actions, I realize that directly moving cubes to their target panels without violating any rules is our primary goal.

Bob can assist by moving the blue_square from panel5 to panel3, which is within both of our reach ranges. This action is crucial as it does not violate any forbidden actions; it involves moving a cube to an intersection panel (panel3) to facilitate its transfer between zones. After Bob moves the blue_square to panel3, I can then pick it up and place it on its target panel, panel2, in a subsequent round.

Bob's task of placing the pink_polygon on panel4 does not require immediate action since it's already in the correct position. Chad's task involves the yellow_trapezoid, which is also correctly placed on panel6. Therefore, Chad does not need to take any action regarding the yellow_trapezoid this round.

Given the current situation, my proposal for this round is for Bob to assist by moving the blue_square closer to me, while I wait for the blue_square to be within my reach. Chad can also wait since his cube is already in the correct position.

Let's confirm if this plan works for everyone. If there are no objections or further suggestions, we can proceed with the execution.

**PROCEED**

### Bob - Round 2 (Conclusion)

Since my target is to place the pink_polygon on panel4, and other robots have their respective targets with Alice needing to place the blue_square on panel2 and Chad needing to place the yellow_trapezoid on panel6, we need to coordinate our actions carefully. The pink_polygon is already correctly placed on panel4, which means my task for this round does not require any action regarding the pink_polygon.

Alice has proposed that I assist by moving the blue_square from panel5 to panel3 to bring it within her reach. This action is within my capabilities and does not violate any forbidden actions, as it involves moving a cube to an intersection panel (panel3) to facilitate its transfer between zones. After moving the blue_square to panel3, Alice can then pick it up and place it on its target panel, panel2, in a subsequent round.

Chad's task involves the yellow_trapezoid, which is already correctly placed on panel6. Therefore, Chad does not need to take any action regarding the yellow_trapezoid this round and can wait.

Given the current situation and the proposed actions, my action for this round will be to move the blue_square from panel5 to panel3 to assist Alice. This will allow us to progress towards completing our overall task more efficiently. Since my pink_polygon is already in the correct position, and Chad's yellow_trapezoid is also correctly placed, there are no further actions required from me or Chad regarding our respective cubes.

Let's confirm if this plan works for everyone. If there are no objections or further suggestions, we can proceed with the execution.

**EXECUTE**
NAME Alice ACTION WAIT
NAME Bob ACTION PICK blue_square PLACE panel3
NAME Chad ACTION WAIT

## J.3 ACTION CHECKER PROMPT AND RESPONSE EXAMPLE

We present the prompt and response example of the action checker in a state action pair of the Sandwich domain.

Action Checker 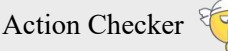

You are an action checker. You should check if the [Action] is valid based on the given [State].

**[Checker Instruction]**
You should follow the check instructions in [Detailed Constraints for Action] to check the [Action] step-by-step. Take the mistakes in the [Action Mistakes] into consideration.

The following lists the [Task Description] and [Action Space Definition].
**[Task Description]**
Task: Cooperative Cooking in a Multi-Agent System

Agents: Chad and Dave
Chad - Can only PICK food items from the right side of the table.
Dave - Can only PICK food items from the left side of the table.

Objective: Collaboratively prepare a meal named "[bacon_sandwich]". The food items of target state must be assembled in the following sequence: bread_slice1, tomato, cheese, bacon, bread_slice2.

**[Action Space Definition]**
1) PICK <obj>: Pick one food <item>. Only PICK if gripper is empty. PICK only the correct next item according to the recipe
2) PUT <obj1> <obj2>: PUT <obj1> on top of <obj2>. <obj1> can be food, <obj2> can be food or cutting_board
3) WAIT, do nothing
Only one robot can PUT each round. You must PICK up an item before PUT
[Action Output Instruction]
Must first output 'EXECUTE\n', then give exactly one action per robot, put each on a new line
Dave can only pick up the food item on the left side of the table. Chad can only pick up the food item on the right side of the table

**[Detailed Constraints for Action]**
Check the following constraints and Fill in blanks in '[]'. Check these constraints one by one: 1, 2...
* Get [Extended Recipe Order]: [cutting_board, bread_slice1, ...] (cutting_board + [Recipe Order])
1. [Line Number Check] The line of the actions should be less than or equal to two.
2. [Agent Order Check] The first agent should be Chad and the second agent should be Dave.
3. [WAIT Number Check] The number of WAIT should be less than or equal to one.
4. [PUT Number Check] The number of PUT should be less than or equal to one.
5. [PICK Side Check] PICK the food item on the side of the robot.
6. [PICK Robot State Check] PICK the food item only if the gripper is empty.
7. [PICK Food State Check] PICK the food item only if the food item's state is not 'atop <obj>'.
8. [PUT Availability Check 1] If <obj2> is cutting_board, <obj1> must be bread_slice1.
9. [PUT Availability Check 2] If <obj2> is not cutting_board, state of <obj2> should equal to the food item before <obj2> in the [Extended Recipe Order].
10. [PUT Availability Check 3] If no other food item is 'atop <obj2>', this constraint is valid directly.
11. [PUT Order Check 1] <obj1> should equal to the first item in [Reachable Items].
12. [PUT Order Check 2] <obj2> should equal to the food item before <obj1> in [Extended Recipe Order].
13. [PICK Order Check] For PICK action, robot should PICK the first food item in its [Reachable Items].

**[Action Mistakes]**
No action mistakes.

Now you should check the following [Action] based on the given [State]
[State]
[Food States]
bread_slice1: atop cutting_board
bread_slice2: gripped by Dave
bacon: atop cheese
cheese: atop tomato
tomato: atop bread_slice1

cucumber: on right side
ham: on right side
beef_patty: on right side
[Robot States]
1. Chad's gripper is empty
2. Dave's gripper is holding bread_slice2

EXECUTE
NAME Chad ACTION WAIT
NAME Dave ACTION PUT bread_slice2 bacon

[Start]
[Extended Recipe Order]: ..., the first item in [Reachable Items]: ...
1.

---

[Extended Recipe Order]: [cutting_board, bread_slice1, tomato, cheese, bacon, bread_slice2].

1. [Line Number Check] The line of the actions should be less than or equal to two: [yes]
2. [Agent Order Check] The first agent should be Chad and the second agent should be Dave: [yes]
3. [WAIT Number Check] The number of WAIT should be less than or equal to one: [yes]
4. [PUT Number Check] The number of PUT should be less than or equal to one: [yes]
5. [PICK Side Check] PICK the food item on the side of the robot: [not applicable]
6. [PICK Robot State Check] PICK the food item only if the gripper is empty: [not applicable]
7. [PICK Food State Check] PICK the food item only if the food item's state is not 'atop <obj>': [not applicable]
8. [PUT Availability Check 1] If <obj2> is cutting_board, <obj1> must be bread_slice1: [not applicable]
9. [PUT Availability Check 2] If <obj2> is not cutting_board, state of <obj2> should equal to the food item before <obj2> in the [Extended Recipe Order]: [yes]
10. [PUT Availability Check 3] Get <item>s in [State] whose state matches '<item>: atop bacon': pool=[]. If no other food item is 'atop bacon' (meaning pool is empty), this constraint is valid directly: [yes]
11. [PUT Order Check 1] <obj1> should equal to the first item in [Reachable Items]: [yes]
12. [PUT Order Check 2] <obj2> should equal to the  food item before <obj1> in [Extended Recipe Order]: [yes]
13. [PICK Order Check] For PICK action, robot should PICK the first food item in its [Reachable Items]: [not applicable]

**[Conclusion]**
The [Action] is valid based on the given [State].

---

The action checker is based on state in both single-agent and multi-agent settings, so the checking process for all tasks is similar.

## J.4 WORLD MODEL PROMPT AND RESPONSE EXAMPLE

We present the prompt and response example of the world model in a state action pair of the BlocksWorld task.

---

**World Model** 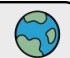

You are a state predictor. You will be given a [State] and an [Action] as input. should predict the [State] after the [Action] is applied to the given [State]. You should follow the [Predicting Instruction] to predict the [State].

**[Predicting Instruction]**
You will be provided with the [State] and the [Action] of the agent. You should think step by step to output the [Prediction] of the next [State] based on the given [State] and [Action]. The format of the [Prediction] should follow the [Detailed Constraints for State].
Please output your thinking process step-by-step by following theses steps:
The most important thing: Follow the instructions step-by-step and ensure each step is completed precisely. Repeat the instructions and fill in the blanks '[]' without introducing any modifications or additional content.
1. [Interaction Item Pool]: initialize the pool of the blocks that the agent is interacting with. It is a empty list at the beginning: pool={}.
2. [Action Forward Rule]: Follow these steps to predict the [Prediction]:
  + If the action is "PICK UP" or "UNSTACK", the state of <object> in the [Prediction] should be "in hand". The state of your hand in the [Prediction] should be "Holding <object>". Add <object> to the pool: pool=[].
  + If the action is "PUT DOWN", the state of <object> in the [Prediction] should be "on table". The state of your hand in the [Prediction] should be "Empty". Add <object> to the pool: pool=[].
  + If the action is "STACK ON", the state of <object> in the [Prediction] should be "on <target>". The state of your hand in the [Prediction] should be "Empty". Add <object> to the pool: pool=[].
3. [Prediction Conclusion]: Conclude the [Prediction] based on the [Action Forward Rule].
  + The format of the [Prediction] should follow the [Detailed Constraints for State].
  + The blocks that are not present in the pool should not change their state.

The following lists the [State Space Definition].
**[State Space Definition]**
Define the state of the scenarios: blocks and hand.
1. [Hand State]: The state of the hand. It can be "Empty" or "Holding <object>", where <object> is a block.
2. [Block States]: The state of each block. It can only be one of ["on <object>", "on table", "in hand"]. The blocks should be listed in this order: blue block, orange block, red block, yellow block.

**[State Template]**
[State]
[Hand State]
<hand state>
[Block States]
blue block: <block state>
orange block: <block state>
red block: <block state>
yellow block: <block state>

**[Detailed Constraints for State]**
The following paragraphs are the detailed constraints for State. There are three main <check items>: [Title Check], [Hand State Check], and [Block States Check].
1. [Title Check] A header titled [State].
2. [Hand State Check] A header titled [Hand State]. Then, list the Hand State.
  + Hand state must be one of ["Empty", "Holding <object>"], where <object> is a block (e.g. "Holding blue block").
3. [Block States Check] A header titled [Block States]. Then, list the states of each block.
  + Each line represents the state of one block item.
  + The block items must be listed in this order: blue block, orange block, red block, yellow block.
  + The <block state> specifies the state of the block. It can only be "on <object>", "on table" or "in hand" (e.g. "blue block: on table", "orange block: on red block", "red block: in hand").

[Prediction Mistakes]
No prediction mistakes.

Now, you should predict the next [State] after the following [Action] is applied to the given [State]. You have to follow the instructions strictly rather than your own thoughts.

---

The response of the world model is relatively concise, as it only needs to infer the next state according to the given instructions.

[State]
[Hand State]
Holding blue block
[Block States]
blue block: in hand
orange block: on yellow block
red block: on orange block
yellow block: on table

EXECUTE
STACK blue block ON red block

---

[Interaction Item Pool]: initialize the pool of the blocks that the agent is interacting with. It is an empty list at the beginning: pool={}.
[Action Forward Rule]: List the action: [STACK blue block ON red block], the state in the [Prediction] is changed by the [Action].
- If the action is "STACK ON", list the <object> of action: [blue block], and <target> of action: [red block]. The state of <object> in the [Prediction] should be changed to "on red block". The state of your hand in the [Prediction] should be changed to "Empty". Add <object> to the pool: pool=[blue block].

[Prediction Conclusion]:
[State]
[Hand State]
Empty
[Block States]
blue block: on red block
orange block: on yellow block
red block: on orange block
yellow block: on table

## J.5 Prediction Checker Prompt and Response Example

We present the prompt and response example of the prediction checker in a state action pair of the Sort domain.

Prediction Checker 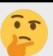

You are a prediction checker. You will receive a [State], an [Action], and the subsequent [Prediction]. You should check if the [Prediction] is valid based on the given [State] and [Action]. You should follow the [Checker Instruction] to check the [Action].

**[Checker Instruction]**
You should follow the instructions in [Detailed Constraints for Prediction] to check if the [Prediction] is valid based on the given [State] and [Action]. Take the mistakes in the [Prediction Mistakes] into consideration.

The following lists the [Detailed Constraints for Prediction].
**[Detailed Constraints for Prediction]**
Check the following constraints and Fill in blanks in '[]'. Check these constraints one by one: 1, 2... Only get your conclusion according to the results of these checks!
* Get [Interaction Item Pool]: list all PICKed objects in the [Action]: <pool>=[].
1. [Header Check] A header titled [State] -> A header titled [Cube States] -> List the Cube States.
2. [Cube Order Check] Cubes must be listed in the following order: [blue_square, pink_polygon, yellow_trapezoid].
3. [Cube Format Check] Each line should be formatted as "<cube_name> is on <panel_name>". Three lines in total.
4. [Cube State Check] Don't have two cubes on the same panel.
5. [PICK & PLACE Rule Check] For PICK & PLACE, state of <object> should be '<object> is on <target>'.
* When checking, you should repeat the prompt and fill in blanks, replace <obj> and <agent> with their true values, and give the line conclusion (yes/no/not applicable). For example:
4. [Cube State Check] Repeat cubes' locations: [panel1, panel2, panel3]. Don't have two cubes on the same panel: [yes]. Line conclusion: [yes]
* Constraint 5 may have multiple agents of checking. If so, you should output results separately. For example:
5.1 [PICK & PLACE Rule Check] For Alice's action: [PICK blue_square PLACE panel2], repeat <object>=[blue_square], <target>=[panel2], state of <object> in the prediction: [blue_square is on panel2], it should be '<object> is on <target>': [yes]. Line conclusion: [yes]
5.2 [PICK & PLACE Rule Check] For Bob's action: ... 5.3 ...

**[Prediction Mistakes]**
No prediction mistakes.

Now you should check the following [Prediction] based on the given [State] and [Action]. You have to check the instructions step-by-step strictly rather than your own thoughts.
**The original state is:**
[State]
[Cube States]
blue_square is on panel6
pink_polygon is on panel4
yellow_trapezoid is on panel2

**The action is:**
EXECUTE
NAME Alice ACTION PICK yellow_trapezoid PLACE panel1
NAME Bob ACTION WAIT
NAME Chad ACTION PICK blue_square PLACE panel5

**The prediction is:**
[State]
[Cube States]
blue_square is on panel5
pink_polygon is on panel4
yellow_trapezoid is on panel1

[Start]
Get [Interaction Item Pool]: list all ...
1. [Header Check]
2. [Cube Order Check] ...
[Conclusion] ...

Due to passing all the checks, this prediction is considered correct.

---

**[Start]**
Get [Interaction Item Pool]: list all PICKed objects in the [Action]: <pool>=[yellow_trapezoid, blue_square].

1. [Header Check] A header titled [State] -> A header titled [Cube States] -> List the Cube States: [yes].
2. [Cube Order Check] Cubes must be listed in the following order: [blue_square, pink_polygon, yellow_trapezoid]: [yes].
3. [Cube Format Check] Each line should be formatted as "<cube_name> is on <panel_name>". Three cube lines in total: [yes].
4. [Cube State Check] Repeat cubes' locations: [panel5, panel4, panel1]. Don't have two cubes on the same panel: [yes]. Line conclusion: [yes].
5. [PICK & PLACE Rule Check]
5.1 For Alice's action: [PICK yellow_trapezoid PLACE panel1], repeat <object>=[yellow_trapezoid], <target>=[panel1], state of <object> in the prediction: [yellow_trapezoid is on panel1], it should be '<object> is on <target>': [yes]. Line conclusion: [yes].
5.2 For Bob's action: [WAIT], no object interaction to check. Line conclusion: [not applicable].
5.3 For Chad's action: [PICK blue_square PLACE panel5], repeat <object>=[blue_square], <target>=[panel5], state of <object> in the prediction: [blue_square is on panel5], it should be '<object> is on <target>': [yes]. Line conclusion: [yes].

**[Conclusion]** All checks are valid based on the provided [Action] and the resulting [Prediction]. The prediction is accurate.

---

