# OpenReview forum: "Empowering LLM Agents with Zero-Shot Optimal Decision-Making through Q-learning"
_ICLR.cc/2025/Conference — ICLR 2025 Poster_

### Official Review · Reviewer_ZK4E · 2024-11-01

**Soundness:** 2
**Presentation:** 2
**Contribution:** 2
**Rating:** 3
**Confidence:** 4

**Summary:**

In this work, the authors develop an algorithm that combines the zero-shot capabilities of large language models (LLMs) with the decision-making strengths of reinforcement learning (RL). They introduce the Model-based LLM Agent with Q-Learning (MLAQ), which leverages Q-learning to derive optimal policies from transitions stored in memory. Unlike traditional RL agents that rely on data collected from direct environmental interactions, MLAQ constructs an "imagination space" based entirely on the LLM to simulate interactions and derive zero-shot policies. The results indicate that MLAQ achieves an optimal success rate of over 90% on tasks where other methods have struggled to perform effectively.

**Strengths:**

- The innovation of the initial idea is good but the works has a lot of engineering! The method needs to be reformed.

**Weaknesses:**

-  The abstraction and introduction is well written but the rest is not fluent.
- LLM-based framework for an agent is the same as to decision transformers.  The author did not talk about that and even compare with this
- The paper is not well-organized! Some terms need to be defined before going to the details of the method.
- Some definitions are not clear: like what is exactly imagination space.
- They said the work does not use environmental tools but they use T which is all information about the environment including actions, states, ...
- There are other type of works which are zero-shot and they paper did not mention them in either related work or introduction: Sub-goal Distillation: A Method to Improve Small Language Agents.

**Questions:**

- What does this mean "the ground-truth available actions δ(s)"? Node selection is not clear.
- The paper claimed for a zeros-shot approach! It is true but at the same time there is a lot of engineering part in the method like correct the imaginary transitions to refine the LLM-based world model, which makes the concept of zero-shot unsatisfied even though it is zero-shot!
- The criteria for the results are not enough! That should be proper comparison with other methods.

---

> ### Author Response · Authors · 2024-11-20
> **Authors Response (Part 1/2)**
>
> **Q1: What are the differences between MLAQ and decision-transformer?**
>
> **A1:** Decision Transformer and LLM, especially LLM agents, are **fundamentally different approaches**.
>
> **Similarity:** Both decision-transformer and LLM use transformer architecture and autoregressive computation, requiring large amounts of offline data for pretraining.
>
> **Differences**
>
> + **Input and Output:** Decision-transformer requires inputs such as state, action, and return-to-go, **at the tensor level**. Each input type is mapped to features using specific embedding layers, with outputs are only for actions. In contrast, LLM (and agents) operates at the token level, converting natural language words or characters into tokens through a large embedding layer, with output as a probability distribution **over all tokens**.
> + **Optimization:** Decision-transformer relies on a large amount of expert data for training and can only solve tasks present in the dataset. The decision-making capability of LLM agents primarily derives from the **general comprehension capability acquired through pretraining on human data**, without reliance on expert in decision-making for supervised learning.
>
> ---
>
> **Q2: What are environmental tools, and does MLAQ use them?**
>
> **A2:** Environmental tools refer to **functions provided by the environment**, such as the dynamics function, which predicts the next state after executing an action, and the available action function, which lists all available actions in the current state. MLAQ **does not use any environmental tools**, and achieve these by the proposed MLAQ framework.
>
> **Environmental tools trade-off generalization for performance, while MLAQ could maintain efficiency through domain description "T" (actually is \tau).** The domain description, analogous to a game manual, provides only key information such as state and action space rather than "all information" (humans also need instructions or tutorials to play games). Experimental results show that the current LLM has sufficient capability to **achieve the available actions exploration and next-state predictions**.
>
> ---
>
> **Q3: What is the ground-truth available actions?**
>
> **A3:** The ground-truth available actions represent **the set of actions an agent can choose** to interact with the environment in a given state.
>
> **Available actions:** Taking the Sort task in RoCo as an example, an agent can select to "pick one of three objects and place it on a panel". But not all actions are available, e.g., picking objects outside the agent's reach range are considered unavailable actions.
>
> Methods like RAP obtain ground-truth available actions using environmental tools, but **MLAQ explore them from scratch**. During the node selection phase, each state node begins with an empty available action set. MLAQ alternates between exploiting existing child nodes and exploring new available actions (and corresponding child nodes), progressively expanding the available action set, denoted as \delta(s).
>
> ---
>
> **Q4: Is MLAQ a zero-shot method with a lot of engineering part?**
>
> **A4:** MLAQ is a zero-shot method, and it is a general algorithm rather than an engineering solution tailored to some specific scenarios.
>
> **Zero-shot:** We found that there are some ambiguities in definitions, with zero-shot having **two different definitions**: making decisions without providing examples in LLM, and making optimal decisions without environmental interaction. Although the starting point of our study is the second goal, MLAQ actually **satisfies both definitions**. Furthermore, MLAQ requires feedback from the environment only when decision errors occur (MLAQ becomes few-shot), but the "Env replans" metric in Table 2 is **very close to zero**, which indicates that MLAQ achieves zero-shot decision-making in most cases.
>
> **Engineering part:** MLAQ is a general LLM-based agent framework. To apply it in other domains, users only need to provide a domain description in natural language as a manual, without the need for environmental tools. For example, in gaming scenarios, **users only need to provide the game tutorial (a manual) rather than source codes**.

---

> ### Author Response · Authors · 2024-11-20
> **Authors Response (Part 2/2)**
>
> **Citations of more related works about zero-shot planner**
>
> Although the paper mentioned by the reviewer explicitly indicates an approach for small language models in the title and does not mention zero-shot at all, we have cited it in the related work section of the revised manuscript and added references on zero-shot planners, including:
>
> 1.  Huang, Wenlong, et al. "Language models as zero-shot planners: Extracting actionable knowledge for embodied agents." ICML, 2022.
>
> 2. Hashemzadeh, Maryam, et al. "Sub-goal Distillation: A Method to Improve Small Language Agents." CoLLAs, 2024.
>
> 3. Gkanatsios, Nikolaos, et al. "Energy-based Models are Zero-Shot Planners for Compositional Scene Rearrangement." RSS, 2023.
>
> 4. Kwon, Teyun, Norman Di Palo, and Edward Johns. "Language models as zero-shot trajectory generators." IEEE RAL, 2024.
>
> 5. Wang, Lei, et al. "Plan-and-Solve Prompting: Improving Zero-Shot Chain-of-Thought Reasoning by Large Language Models." ACL, 2023.
>
> ---
>
> **More comparisons**
>
> We compared our method with many LLM agents on BlocksWorld and with the original RoCo algorithm on the RoCo-benchmark. Some algorithms (e.g., ReAd) have been tested on the RoCo-benchmark using evaluation methods different from ours. We commit to comparing our approach with theirs as soon as their code is open-source. Furthermore, we **conduct an experiment on Crafter (please refer to the overall response for more detail)** and compare our method with multiple LLM agents and RL agents to demonstrate its effectiveness.

---

> ### Author Response · Authors · 2024-11-24
> **Kindly Inquire Whether We Have Addressed Your Concerns**
>
> Dear Reviewer,
>
> **Thank you** once again for taking the time to review our manuscript and for considering our responses to your comments.
>
> If you require any further clarification or would like additional points to be included in our response, we welcome any further discussion to ensure everything is clear and satisfactory. If we have adequately addressed your concerns, we would sincerely appreciate your **re-evaluation** and **reconsideration** of your rating.
>
> Thank you for your consideration.
>
> Best regards,
>
> Authors

---

> > ### Comment · Reviewer_ZK4E · 2024-11-25
> >
> > - I went through the response. I will keep my score
> > - My concerns still remain about the claim of the paper for zero-shot and a lot of manual handling of each part of the algorithm.
> > - Also as I mentioned it is not well written and has lack of clarity.

---

> > > ### Author Response · Authors · 2024-11-29
> > > **Looking Forward to More Feedback from the Reviewer**
> > >
> > > Dear Reviewer,
> > >
> > > We're eagerly awaiting the reviewer's insightful feedback. Despite the fact that we can't submit a revised version now, we pledge to incorporate the conclusions from our discussions into the final manuscript. We sincerely hope that the reviewer can point out the unclear points more specifically to help us explain to the reviewer more targetedly.
> > >
> > > Sincerely,
> > >
> > > Authors

---

> ### Author Response · Authors · 2024-11-25
>
> Dear Reviewer ZK4E,
>
> We still sincerely thank you for your time to review our response. We would be grateful if you could provide more feedback.
>
> ---
>
> **Claim of the paper**
>
> + **Zero-shot**. We mentioned **two definitions** of zero-shot in the response, and explained that MLAQ meets these two definitions, but indeed in a very few cases it will need feedback from the environment, so **we have been emphasizing in the manuscript that it is zero- or few-shot**. We sincerely want to know which part of this claim does the reviewer specifically disagree with?
>
> + **Manual handling**. We understand that the reviewer's opinion may come from the different research background, so **we sincerely want to clarify that this is our general algorithm design, which does not introduce overly complex modules compared to traditional model-based RL**.
>
>   + The main framework of MLAQ is **in the style of Dreamer**, which is one of the most famous model-based RL methods. In order to adapt to LLM-based decision-making, we **replace the world model** that needs task-specific data training with a more general LLM. We also propose **a novel planning approach** that enables the LLMs to explore the available actions from scratch, and **a mixed-examination** to help the LLM-based world model approach the true MDP.
>
>   + So we sincerely want to know **what the reviewer specifically thinks are the "manual handling parts"**? Is our Dreamer-style model-based RL framework too complex, or something else?
>
> ---
>
> **Writing**
>
> We **sincerely hope** to receive feedback from the reviewer on **where exactly is not well-written**, and **which points are still unclear** to the reviewer ZK4E?
>
> Since **Reviewer ouUC, 4yYK, and FSC6 all think that this paper is well-written**, the feedback from the Reviewer ZK4E **is critical for us to revise our manuscript** when the readers do not have more patience to understand this paper.   :)
>
> ---
>
> **Sincerely**,
>
> Authors

---

### Official Review · Reviewer_XLz6 · 2024-11-04

**Soundness:** 2
**Presentation:** 1
**Contribution:** 2
**Rating:** 3
**Confidence:** 3

**Summary:**

This paper considers the problem of enabling LLM agents to complete long-horizon tasks. To accomplish this, the paper proposes a method that combines Q-learning, a way to generate imaginary data using LLMs, an MCTS-style planner that acts in that space, a mechanism to improve the quality of the imaginary data, and a variant of UCB that facilitates exploration.

**Strengths:**

- Addresses an interesting problem of enabling LLMs to solve long-horizon tasks
- Strong performance on prior benchmarks
- Comparisons to relevant prior works
- Ablations confirm the importance of various components in BlocksWorld

**Weaknesses:**

Overall, the clarity of the writing prevents me from recommending this paper for acceptance. I tried reading parts of the paper several times, but I unfortunately still don't understand truly how the method works. The writing seems both hard to follow/understand and missing important information. (I have published papers in the realm of LLMs, LLM agents, Q-learning/reinforcement learning, and model-based RL, so I should have sufficient background to understand the submission.)

It's hard to go through every part of the paper and describe what is hard to follow or missing, but here are a few examples:
- In the intro: It’s not clear what problem statement this paper is aiming to address. The sentences do not have a clear train-of-thought, and concepts are introduced without fully explaining them / without giving concrete examples.
- Important details are missing: how is the Q-function represented? The preliminaries describe a tabular representation, but this doesn’t seem applicable to the language setting. Is there a policy improvement step using the Q-function? If so, how is it done? How do each of the components interact? What concretely is the action space being considered? What is the breadth of the actions that the LLM can take?
- The term “environmental tools” is used extensively, but it’s not entirely clear what it means. It hasn't been used in prior papers to my knowledge, and isn't described clearly in this paper. It reminds me of LLM "tools", but based on the brief description in the intro, this seems incorrect and it seems to more be referring to assumptions that the method makes.
- The paper is not self-contained, and assumes detailed knowledge of specific prior works, e.g. Mandi et al to understand the RoCo task.
- The figures are helpful, but quite abstract
- The abstract refers to a website (https://mlaq.site/) that is empty
- Qualitative examples of the agent performance would be helpful
- Which LLM is used in the experiments? How large is it?


Secondarily, the method is very complex involving many different components. This fact combined with the lack of clarity in the paper means that it would be impossible to implement the method based on the description in the paper. I appreciate the ablation study though.

**Questions:**

see weaknesses

---

> ### Author Response · Authors · 2024-11-20
> **Authors Response**
>
> Dear Reviewer, thank you for your time to read our paper and provide useful comments. We carefully address your concerns as follows. Should our revisions adequately address your concerns, we would be most grateful for your positive re-evaluation.
>
> ---
>
> **Q1: What is the problem statement of this paper?**
>
> **A1:** This paper aims to address the challenge of empowering LLM agents with **zero- or few-shot optimal decision-making capabilities**.
>
> MLAQ is a general algorithm designed for decision-making tasks. Leveraging terminology from the LLM reasoning domain, MLAQ is a **test-time compute method in LLM-based decision-making**, further enhancing the optimal decision-making capability of LLM-based basic policies through Q-learning in LLM-based imaginary transitions.
>
> ---
>
> **Q2: What is the environmental tools?**
>
> **A2:** Environmental tools refer to **functions from the environment** to enhance the LLM agents, such as using the environment's dynamics function for next-state prediction, available action function to obtain ground-truth available actions, and reward function for feedback on decisions. The reviewer correctly pointed out the similarity between "environmental tools" and LLM "tools". **This is similar to an LLM agent using a calculator as a tool for mathematical computations.**
>
> ---
>
> **Q3: how is the Q-function represented and is there a policy improvement step using the Q-function?**
>
> **A3:** The Q-function in MLAQ is in a tabular form, but it is **not represented as a matrix**, since the ground-truth available actions are unknown. Besides, MLAQ does not have a policy improvement step, and it directly uses the argmax actions supported by the Q-value function to make decisions.
>
> + Get the Q-value: For a given state, the Q-function stores its available actions and corresponding Q-values. These available actions are explored through UCB-guided planning in the imagination space, so they **may not cover all ground-truth available actions**. Therefore, we only output the argmax action from the **explored available actions**.
> + The state space in tasks based on natural language is typically small, as the problem is inherently abstracted into natural language. Even for tasks like Crafter, the overhead of storing transitions using a tabular approach is **relatively low**. For future large-scale tasks, MLAQ can efficiently extend the current tabular Q-function to a neural network form using embedding networks like SentenceBert.
>
> ---
>
> **Q4: What concretely is the action space being considered?**
>
> **A4:** The output dimensions is equal to **the number of explored available action** under the given state.
>
> The action space in this paper refers to the description of all actions in the domain description (such as pick or put, regardless of whether they are available or not). We believe the reviewer was actually asking about the **Q-function's output dimension**.
>
> The potential actions for LLM-based basic policy is extremely huge, requiring **token-level combinations**, whereas MLAQ does not use any environmental tools to obtain ground-truth available actions. To this end, we propose a novel UCB variant that supports planning under an **unknown available action set**, enabling iterative exploration of all available actions from scratch. While the breadth of actions LLMs can take might appear infinite, we leverage an LLM-based action checker to retain only actions identified available by the LLM, thereby preventing unbounded expansion.
>
> ---
>
> **Q5: Is the website available?**
>
> **A5:** The correct link to the website is http://mlaq.site/ (HTTP rather than HTTPS), and the GitHub link for the open-source code is https://github.com/laq2024/MLAQ/tree/main. If the reviewer is unable to access the website, please try the original link: http://27.106.114.122:8501/.
>
> ---
>
> **Q6: What is the LLM used in MLAQ?**
>
> **A6:** The LLM used is GPT-4-0125-preview, with other hyperparameters also listed in Table 9 of the previous manuscript.
>
> ---
>
> **Overall response**
>
> This paper aims to introduce the model-based RL approach into LLM agents for **zero- or few-shot optimal decision-making**. Given the limited space, it strives to present relevant concepts to potential readers from both fields, which may lead to some imbalance in the coverage of some concepts. As a result, while some reviewers may find the paper **well-written**, others might feel the explanations lack clarity. We **sincerely appreciate** the reviewers' feedback on areas of confusion, which have been invaluable in improving the manuscript.
>
> The detailed explanations regarding the reviewers' concerns are primarily located in the abstract and introduction sections. In the revised version, we have addressed all concerns raised by the reviewers and provided **more definitions and explanations for key terms** to ensure accessibility for readers from different backgrounds.

---

> ### Author Response · Authors · 2024-11-20
> **Summary of MLAQ**
>
> **Motivation:** This paper combines the general comprehension capability of LLMs and the optimal decision-making ability of RL to achieve zero- or few-shot optimal decision-making.
>
> ---
>
> **Contributions**
>
> **Summary:** The LLM-based basic policy and world model leverage the general comprehension capability of LLMs to generate imaginary transitions without interacting with the environment. Q-learning is used on transitions stored in memory to achieve optimal decision-making.
>
> **LLM-based Imagination Space:** The basic policy outputs possible actions given a state, while the world model predicts the next state based on state-action pairs, generating imaginary transitions in a Dreamer-style for policy optimization.
> + LLM-based imaginary transitions are generated for Q-function training without environmental tools. Considering the reviewer's background in model-based RL, **this can be likened to Dreamer**. However, we emphasize that LLM’s general capability allows it to replace RSSM-based world model.
> + To address the hallucination issue, our LLM-based self-examination improves the accuracy of imaginary transitions, and env-examination uses ground-truth transition to correct the previously undetected errors.
>
> **Planning in the Imagination Space:** A novel planning approach, driven by our UCB variant, is proposed to enable exploration in LLM-based imagination without ground-truth available actions. The regret bound for this approach is derived theoretically.
> + Our planning approach enables iteratively exploration of available actions without environmental prior knowledge, enhancing the algorithm's generality.
> + Exploration aims to discover potentially better decisions, and the resulting imaginary transitions are used to enhance the agent's decision-making performance via Q-learning. If the basic policy already provides optimal actions, MLAQ does not explore unnecessarily (though users can opt for continuous exploration).
>
> ---
>
> **Main Conclusions**
>
> LLMs can build a model-based framework, achieving performance far exceeding current LLM agents with minimal computational cost, fulfilling the motivation for zero- or few-shot optimal decision-making.

---

> ### Author Response · Authors · 2024-11-24
> **Kindly Inquire Whether We Have Addressed Your Concerns**
>
> Dear Reviewer,
>
> **Thank you** once again for taking the time to review our manuscript and for considering our responses to your comments.
>
> If you require any further clarification or would like additional points to be included in our response, we welcome any further discussion to ensure everything is clear and satisfactory. If we have adequately addressed your concerns, we would sincerely appreciate your **re-evaluation** and **reconsideration** of your rating.
>
> Thank you for your consideration.
>
> Best regards,
>
> Authors

---

> > ### Comment · Reviewer_XLz6 · 2024-11-24
> > **Reply**
> >
> > I looked through the revised paper. My concerns about clarity remain, and I will keep my score.

---

> > > ### Author Response · Authors · 2024-11-25
> > >
> > > Dear Reviewer,
> > >
> > > Regardless of anything, we still want to thank you for taking the time to read our response.
> > >
> > > We are going to provide a further revised manuscript, but before that, we sincerely want to ask **if the explanations in our response have addressed your concerns**. If so, we would like to revise our manuscript **accordingly**; otherwise, we sincerely **ask for your feedback on any points that are still unclear**.
> > >
> > > Sincerely,
> > >
> > > Authors

---

> > > ### Author Response · Authors · 2024-11-29
> > > **Looking Forward to More Feedback from the Reviewer**
> > >
> > > Dear Reviewer,
> > >
> > > We're eagerly awaiting the reviewer's insightful feedback. Despite the fact that we can't submit a revised version now, we pledge to incorporate the conclusions from our discussions into the final manuscript. We sincerely hope that the reviewer can point out the unclear points more specifically to help us explain to the reviewer more targetedly.
> > >
> > > Sincerely,
> > >
> > > Authors

---

> ### Author Response · Authors · 2024-11-25
>
> Dear Reviewer XLz6,
>
> We really want to understand **why the reviewer cannot understand how MLAQ works**, considering that the reviewer mentioned that he/she has published papers in the field of model-based RL, then the reviewer can fully understand the main framework of this paper.
>
> The main framework of MLAQ is **Dreamer-style (For all readers, Dreamer is one of the most famous model-based RL methods)**, but we have **replaced the world model** that needs task-specific data training with a more general LLM. Furthermore, **in order to adapt to LLM-based decision-making**, we have proposed **a novel planning approach** that enables the LLMs to explore the available actions from scratch, and **a mixed-examination** to help the LLM-based world model approach the true MDP. It **DOES NOT** add overly complex modules compared to the original model-based RL, and we have provided **very detailed pseudo codes**, **open sourced our code**, and even **all experimental data**.
>
> We **sincerely ask** the reviewer to be more patient to review our manuscript and responses once again. Considering that **Reviewer ouUC, 4yYK, and FSC6 all think that this paper is well-written**, so we really need the feedback from reviewer XLz6 to see what description we have missed to make a researcher in model-based RL cannot understand MLAQ.
>
> We have submitted a new revised version, trying to emphasize the points that the reviewer did not understand, and we sincerely ask for your feedback to address your concerns.
>
> **Sincerely**,
>
> Authors

---

### Official Review · Reviewer_FSC6 · 2024-11-04

**Soundness:** 3
**Presentation:** 3
**Contribution:** 3
**Rating:** 8
**Confidence:** 3

**Summary:**

In this paper, the authors present a Model-based LLM Agent with Q-learning (MLAQ), which employs Q-learning to derive optimal policies from transitions within memory. Unlike classic RL agents that collect data via environment interactions, MLAG leverages LLM to perform imaginary interactions where LLM can be served or viewed as a world model for the environment simulation. This allows the policies to learn to optimize under the imaginary space. Further, the authors employ a mixed-examination mechanism that utilizes environmental interaction and LLM-based self-examine to enhance the quality of imaginary success. Empirical experiments show that the method can significantly improve the LLM agent performance on several benchmarks, showing promising results.

**Strengths:**

- The idea of using LLM to simulate world models is novel, and the authors give a detailed description of how to leverage LLMs to construct the model to simulate the environment;
- The verification part can guarantee that the model is more accurate than naively simulating the trajectories;

- Empirical experiments show that the idea can achieve significantly better performance, especially for long trajectories;

**Weaknesses:**

- The idea is only evaluated on a very toy game domain. It would be good to see and discuss whether the method can be extended to more complex agent benchmarks, such as function calling, tau-bench, etc.

**Questions:**

- Please see my questions in the weakness session

---

> ### Author Response · Authors · 2024-11-20
> **Authors Response**
>
> Dear Reviewer, thank you for your time to read our paper and provide useful comments. We carefully address your concerns as follows:
>
> **Q: Can MLAQ be extended to more complex scenarios?**
>
> **A:** Yes. To evaluate MLAQ's performance in more complex scenarios, we conduct experiments on Crafter (a 2D version of MineCraft) as our experimental environment. **Please refer to the overall response for more detail.** Based on our experimental results, MLAQ can be demonstrated to have the opportunity to be extended to more complex stochastic problems, but it performs optimally in deterministic, fully observable environments.
>
> + Deterministic, full observability means that LLMs can access complete state information and predict a unique next state based on state-action pairs. Our experiments in BlocksWorld and RoCo demonstrate that LLMs possess this capability, and **MLAQ can leverage imaginary transitions to enhance LLM agents' optimal decision-making** through test-time computation.
> + Results in Crafter experiments indicate that MLAQ's performance decreases when facing environmental stochasticity. We conduct three experiments where the **only difference** lies in how the world model predict next observations **for move actions**, while all other modules **remain consistent** with the original MLAQ. MLAQ-gt, which uses ground-truth environment for state prediction under move actions, outperforms both script-based MLAQ-script and purely LLM-based MLAQ.
>   + On the one hand, MLAQ is **difficult to *adequately explore under limited test-time computation budget** when facing stochastic environments, and the accumulated errors caused by stochasticity significantly impact estimation of Q-values.
>   + On the other hand, current LLMs **preforms poor in modeling stochastic environments**, which further degrades the quality of imaginary transitions and consequently undermines MLAQ's performance. **However, the performance of MLAQ is still better than other existing LLM agents.**

---

> ### Author Response · Authors · 2024-11-24
> **Kindly Inquire Whether We Have Addressed Your Concerns**
>
> Dear Reviewer,
>
> **Thank you** once again for taking the time to review our manuscript and for considering our responses to your comments.
>
> If you require any further clarification or would like additional points to be included in our response, we welcome any further discussion to ensure everything is clear and satisfactory. If we have adequately addressed your concerns, we would sincerely appreciate your **re-evaluation** and **reconsideration** of your rating.
>
> Thank you for your consideration.
>
> Best regards,
>
> Authors

---

### Official Review · Reviewer_4yYK · 2024-11-05

**Soundness:** 2
**Presentation:** 3
**Contribution:** 2
**Rating:** 5
**Confidence:** 3

**Summary:**

The paper presents a novel approach for integrating Q-learning with Large Language Models (LLMs) to enable zero-shot optimal decision-making. The authors propose a MLAQ framework, which includes Q-planner, a memory module, and an imagination space. The model leverage LLM as world model for agent to optimize its decision-making using Q-learning without relying on environmental interactions. The framwork also recruits a Mixed-Examination mechanism for transition verification. Finally, the experiments results outperforms existing methods in optimal decision-making tasks.

**Strengths:**

1. The paper claims an important point of "making optimal decisions" and is well-written.

2. The authors claim that this model is the first work to perform complete RL-based optimization process based on MDP scenario. The idea of generating imaginary transitions with LLMs for Q-learning optimize the policy makes sense to me.

3. The experimental results are good in both tasks.

**Weaknesses:**

1. My biggest concern is about the cost of this framework. While the paper claims that the MLAQ framework can operate without direct interaction with environmental tools by leveraging an LLM-based imagination space, this approach may still incur significant computational and financial costs. LLM interactions, especially those involving step-by-step task simulations, are resource-intensive in terms of both token usage and processing time. Additionally, in ablation experiments, tasks with 8 steps in BlocksWorld were shown to consume at least 40k tokens, which may be expensive for scaling up to complex tasks or multi-agent scenarios. The reliance on LLMs for generating and validating transitions could offset the savings from not interacting with real environments, especially if these transitions require multiple iterations for accuracy.

2. Although MLAQ aims to minimize reliance on real-world data, it still requires occasional feedback from the environment to correct inaccuracies in LLM-generated transitions. In Algorithm 1, it seems the stepwise reward is still from the environment. Then, how does the model deal with sparse or delayed reward situation, e.g. in Figure 3 you can only get binary reward at the last step? In this way, the LLM world model is only used for simulate success trajectories or also provide reward?

3. The LLM-based world model is designed to act as a proxy for real environmental interactions, but this approach may have inherent limitations, especially for tasks requiring high-fidelity simulations or complex physical dynamics. To be more specific, LLMs understanding of actions and outcomes might not always align with real-world behaviors. Does author have more analysis on this?

**Questions:**

1. Since the LLM here serves as a model-based simulator of the real environment and the policy is generated by the Q-function, is it appropreiate to call this "LLM agent"? The action or policy is not diorectly generated by LLM, could this framework be more accurately described as a Q-learning agent that uses an LLM-based world model?

2. I'm still unclear about what the training data is for Q-planner. Is it trained on the synthetic step-wise reward or the ground truth trajectories which is from the interaction with environment?

**Details Of Ethics Concerns:**

The webpage of this paper includes a GitHub link with a non-anonymized account.

---

> ### Author Response · Authors · 2024-11-20
> **Authors Response (Part 1/2)**
>
> Dear Reviewer, thank you for your time to read our paper and provide useful comments. We carefully address your concerns as follows:
>
> ---
>
> **Q1: What are the computational and token costs of MLAQ?**
>
> **A1:** Although MLAQ requires exploration to find more optimal decisions, it **consumes fewer tokens than existing methods while achieving better performance**. MLAQ is a test-time compute method in LLM-based decision-making that combines planning with model-based RL.
>
> **Computational cost.** This mainly involves MLAQ's planning in imagination space. MLAQ's planning in the imagination space is computationally efficient due to its **UCB variant and transition re-utilization**.
> + **Re-utilization of imagination results.** 1) MLAQ re-utilizes transitions in memory during planning to avoid redundant imagination. 2) MLAQ interacts with the environment by using the Q-learning-derived argmax actions, eliminating the need to query the LLM per step.
> + **Efficiency.** MLAQ is an efficient trade-off between basic policy and MCTS. It avoids unnecessary exploration when the base policy is already optimal, **matching traditional agents' costs**. For weaker policies, it uses UCB-guided **exploration to improve performance**. Compared to MCTS (in RAP), MLAQ is more similar to Dreamer, which uses Q-learning for computing state values.
>
> **Token cost.** Due to the transition re-utilization and higher decision correctness, MLAQ has lower token costs than traditional LLM agents.
> + **Bigger memory, then lower token cost.** The 40k token consumption in 8-steps BlockWorld is the average token per episode. In 12-steps, it is decreased to about 25k while the optimal rate significantly outperforming other methods, demonstrating MLAQ's effectiveness.
> + **Lower token cost, but better performance.** Table 2 compares token consumption under RoCo-benchmark, showing that MLAQ uses significantly fewer tokens but achieves higher optimal rate. Unlike RoCo, which requires multiple reflections, MLAQ could output optimal actions even without querying the LLM (Figure 3).
>
> **Why do MLAQ aim to minimize environmental interactions?**
> + We believe that zero-shot optimal decision-making enables LLM agents to **generalize across more scenarios**, requiring neither accurate environmental information (such as source code) nor extensive environment interactions for RL training.
> + Environmental interactions could be costly and raise safety concerns (such as robotic arm operations in real-world), while MLAQ constructs an LLM-based imagination space to derive an optimal policy.
>
> ---
>
> **Q2: How is the reward calculated in MLAQ?**
>
> **A2:** MLAQ achieves sparse rewards by determining whether the next state is equal to the target state and providing positive feedback only when they are equal.
>
> **Clarification of reward.** In Figure 3, the world model is **only** used to predict the next state, not for reward prediction. Besides, the rewards in Algorithm 1 is obtained through environmental interactions, and we also show how sparse rewards are calculated during LLM-based imagination in Algorithm 2.
>
> **Other type of reward function is also available.** We use sparse reward mainly to avoid accessing the ground-truth reward function, but MLAQ also support other reward settings. The experiment on Crafter adopt a reward function with multiple sub-goals, offering positive feedback upon completing sub-goals. Although untested, MLAQ should also have the potential to support dense rewards.
>
> ---
>
> **Q3: How does MLAQ handle situations where environmental complexity exceeds the capability of LLMs?**
>
> **A3:** MLAQ leverages both **LLM's self-examination** to improve transition accuracy and **environmental feedback** to rectify LLMs' output.
>
> However, when the environment becomes **much more complex** (e.g. Crafter), although MLAQ can still outperforms existing LLM agents, using the LLM-based world model performs worse than directly using the ground-truth dynamics function (MLAQ-gt, please refer to the **overall response** for more detail about our experiment on Crafter).
>
> **LLM's self-examination ability has "thresholds".** Current LLMs could fully understand deterministic environments like BlocksWorld, achieving high transition accuracy with LLM-based checkers, as detailed in Sections 4.4 and 4.5. However, LLMs face limitations in more complex environments such as Crafter, resulting in reduced examining accuracy.

---

> ### Author Response · Authors · 2024-11-20
> **Authors Response (Part 2/2)**
>
> **Q4: Should MLAQ be called an LLM agent rather than a Q-learning agent?**
>
> **A4:** We believe the core of MLAQ still remains an LLM agent. MLAQ enhances the optimal decision-making capability of the basic policy through techniques in model-based RL and planning. We would like to maintain its original name rather than changing it to a Q-learning agent, but we greatly appreciate the reviewer's kind suggestions.
>
> **Q5: How to train a Q-planner?**
>
> **A5:** The Q-planner is trained using tabular Q-learning based on transitions in the memory.
>
> We adopt a training framework similar to Dreamer, where transitions are **mainly generated in the LLM-based imagination space**, with environmental transitions being stored **only** when MLAQ makes incorrect decisions. The reward signal is obtained in a sparse reward way for both enironmental and imaginary interactions.

---

> ### Author Response · Authors · 2024-11-24
> **Kindly Inquire Whether We Have Addressed Your Concerns**
>
> Dear Reviewer,
>
> **Thank you** once again for taking the time to review our manuscript and for considering our responses to your comments.
>
> If you require any further clarification or would like additional points to be included in our response, we welcome any further discussion to ensure everything is clear and satisfactory. If we have adequately addressed your concerns, we would sincerely appreciate your **re-evaluation** and **reconsideration** of your rating.
>
> Thank you for your consideration.
>
> Best regards,
>
> Authors

---

> > ### Comment · Reviewer_4yYK · 2024-11-27
> >
> > Thanks to the authors for the rebuttal. The responses provide more clarification of this work. I have read all the reviews and rebuttal from all reviewers. I still have some questions.
> >
> > As authors mentioned Dreamer a lot in the rebuttal, I still feel there are a lot difference between the two works. Dreamer utilized their trained RL agent for exploration. However, in this work, the authors used "LLM-based basic policy". I'm wondering if this imaginary step is related to the learned RL agent. Can you explain more about this?
> >
> > Also, in the imaginary step, it seems the framework has both a LLM-based policy and a LLM world model. How does the world model update? Or, how does the interaction between the real environment improve the LLM based world model and policy?
> >
> > Additonally, since the framwork use LLM to generate action, how does this adapt to continuous space tasks?

---

> > > ### Author Response · Authors · 2024-11-27
> > > **Authors Response (Part 1/2)**
> > >
> > > Dear Reviewer,
> > >
> > > First of all, we **sincerely appreciate** the reviewer for taking the time to read all of our responses. This is a **HUGE** encouragement to us, and we are very much looking forward to **having further discussions** with you and other reviewers (but so far we have not had the opportunity to release this desire for discussion).
> > >
> > > The following are the answers to your questions:
> > >
> > > ---
> > >
> > > **Q1: What is the relationship between MLAQ and Dreamer?**
> > >
> > > **Similarities**: MLAQ and Dreamer are consistent in **motivation and model-based framework**.
> > > + The motivation of both is to **interact with the environment as little as possible**.
> > > + Therefore, we construct an imagination space to **generate imaginary transitions** through the interaction between the world model and the agent (LLM-basic policy in MLAQ). Then, MLAQ and Dreamer use the imaginary transitions to **train the agent (Q-planner in MLAQ)**.
> > >
> > > **Differences**
> > >
> > > + **Dreamer**: In Dreamer, the RL agent that **<interacts with the world model to generate imaginary transitions>** and the one that **<actually makes decisions (and is ultimately evaluated and used) in the environment>** is the same. Dreamer uses a large number of imaginary transitions to **train this RL agent at the parameter level**.
> > > + **MLAQ**
> > >   + **Motivation**: Dreamer still needs a large number of environmental transitions to **train a world model**, and we believe that **the general comprehension capability of LLM can significantly reduce this demand**. Therefore, we introduce an LLM-based basic policy and world model to construct **a fully LLM-based imagination space** without training at the parameter level.
> > >   + **Imaginary step**: Unlike Dreamer, the agent that **<interacts with the world model to generate imaginary transitions>** in MLAQ is an **LLM-based basic policy**. It does not involve RL-based optimization like Dreamer, and **its main role** is to combine with our proposed novel planning approach to achieve **efficient exploration** and generate more **diverse imaginary transitions**.
> > >   + **Environmental step**: The agent (Q Planner) which actually **<makes decisions (and is ultimately evaluated and used) in the environment>** is obtained by using Q-learning for RL-based optimization based on the imaginary transitions in the memory. In this work, we use a **tabular form** to represent the Q function, but inspired by the reviewers, it seems feasible to use a small model to represent the Q-function or policy, which may be implemented in future work.
> > >
> > > **More discussion:** We believe that the main contribution of MLAQ is to demonstrate that **using a model-based RL framework to implement an LLM agent can provide the generalizability of LLM and the optimality of RL**, and we hope to bring some effective information **to the entire community**. To implement an LLM-based model-based framework, we need to solve the shortcomings of LLM in hallucination, optimality, and exploration, so we propose the corresponding technologies:
> > >
> > > 1. **Exploration**: We propose a new **planning approach** specifically for LLM-based decision-making. This approach can efficiently explore when the **ground-truth available actions are unknown**.
> > > 2. **Hallucination**: We introduce a mixed-examination mechanism to enhance the accuracy of imaginary transitions.
> > > 3. **Optimality**: We use **Q-learning** to derive the final agent that interacts with the environment.
> > >
> > > ---
> > >
> > > **Q2: How does the world model update?**
> > >
> > > **A2**: MLAQ **does not train the LLM-based world model and policy at the parameter level**, but starts from **the prompt level**. MLAQ uses env-examination to identify **wrong predictions or unavailable actions based on environmental transitions**, and provides them to the corresponding modules when **encountering the state again** to avoid similar errors.
> > >
> > > The experimental results show that the current LLM has **enough general comprehension capability** to construct a model-based framework without training in the RoCo-benchmark and BlocksWorld used in our experiments. However, in the additional experiment on Crafter conducted during the rebuttal period, LLM **hasn't yet shown the ability to effectively model environments with partial observability and stochasticity**.
> > >
> > > **Discussion**: Although we have not tried, for more complex problems, users can also use environmental transitions to **supervised-finetune the LLM at the parameter level** to improve its accuracy.

---

> > > ### Author Response · Authors · 2024-11-29
> > > **Has Our Response Addressed Your Concerns?**
> > >
> > > Dear Reviewer,
> > >
> > > We have provided more detailed discussions and responses to the reviewer's new questions, and provided our insights into MLAQ and the entire field. We sincerely hope to have in-depth discussions with the reviewer on this.
> > >
> > > If the discussion effectively resolves the reviewer's concerns, we're sincerely hoping for a positive re-evaluation of our work.
> > >
> > > Sincerely,
> > >
> > > Authors

---

> > > ### Author Response · Authors · 2024-12-04
> > > **Has Our Response Addressed Your Concerns? (Once Once Again)**
> > >
> > > Dear Reviewer,
> > >
> > > We have provided more detailed discussions and responses to the reviewer's new questions, and provided our insights into MLAQ and the entire field. We sincerely hope to have in-depth discussions with the reviewer on this.
> > >
> > > If the discussion effectively resolves the reviewer's concerns, we're sincerely hoping for a positive re-evaluation of our work. (Sigh :(
> > >
> > > Sincerely,
> > >
> > > Authors

---

> ### Author Response · Authors · 2024-11-27
> **Authors Response (Part 2/2)**
>
> ---
>
> **Q3: How does MLAQ adapt to continuous space tasks?**
>
> **A3**: The problem mentioned by the reviewer is **a common problem with LLM agents**. At present, almost all LLM agents **cannot** adapt well to continuous space tasks. We have not specifically tried this, so we only **share our views** with the reviewer for reference.
>
> On the one hand, while keeping the LLM network structure unchanged, most of the existing methods try to c**onvert continuous actions into discrete actions**, such as abstracting movement and navigation into natural language instructions in LLM-based embodied agents [1, 2]. On the other hand, some works also try to **introduce some embedding layers on the basis of LLM** to handle different types of states and actions, but this requires additional data for training [3]. We believe this is **still a challenge** for LLM agents where there is significant room for improvement.
>
> [1] Yijun Yang, et al. LLM-Planner: Few-Shot Grounded Planning for Embodied Agents with Large Language Models, ICCV, 2023
>
> [2] Chan Hee Song, et al. Embodied multi-modal agent trained by an llm from a parallel textworld, CVPR, 2024
>
> [3] Jing-Cheng Pang, et al. KALM: Knowledgeable Agents by Offline Reinforcement Learning from Large Language Model Rollouts, NeurIPS 2024
>
> ---
>
> Thanks again for your response!
>
> Sincerely,
>
> Authors

---

> ### Author Response · Authors · 2024-12-02
> **Has Our Response Addressed Your Concerns? (Once Again)**
>
> Dear Reviewer,
>
> We have provided more detailed discussions and responses to the reviewer's new questions, and provided our insights into MLAQ and the entire field. We sincerely hope to have in-depth discussions with the reviewer on this.
>
> If the discussion effectively resolves the reviewer's concerns, we're sincerely hoping for a positive re-evaluation of our work. (Sigh :(
>
> Sincerely,
>
> Authors

---

### Official Review · Reviewer_ouUC · 2024-11-08

**Soundness:** 3
**Presentation:** 3
**Contribution:** 3
**Rating:** 8
**Confidence:** 4

**Summary:**

This paper proposes the Model-based LLM Agent with Q-Learning (MLAQ) to address the gap in optimal decision-making capabilities for large language model (LLM) agents. By combining Q-learning with LLMs, MLAQ achieves zero-shot optimal decision-making by using an LLM-generated imagination space, enabling the agent to derive policies without direct environmental interaction. Notably, the authors introduce a UCB variant to balance exploration and exploitation, and a mixed-examination mechanism to enhance the quality of imaginary data. The framework shows superior performance across single-agent and multi-agent benchmarks (BlocksWorld and RoCo-benchmark), surpassing existing methods in achieving optimal decisions.

**Strengths:**

1. MLAQ presents a novel integration of Q-learning and LLMs, using an LLM-based imagination space to enable optimal decision-making without direct environmental data. This imaginative interaction is a valuable innovation, given the limitations of previous approaches relying heavily on environmental feedback.

2 The UCB variant and mixed-examination mechanisms are technically sound and provide meaningful contributions. These components contribute to reducing computational requirements and minimizing regret, which are crucial in complex decision-making scenarios.

3 The empirical results, showing over 90% optimal success in challenging benchmarks, convincingly demonstrate MLAQ's ability to outperform existing LLM agents. The performance on long-horizon tasks suggests MLAQ’s potential for real-world applications requiring complex decision-making sequences.

4 The proposed MLAQ framework appears modular and adaptable, making it feasible for extension to various domains and applications, including future deployment on physical robotic platforms.

**Weaknesses:**

1. While the technical elements (such as the UCB variant) are well-motivated, the theoretical analysis could be expanded, especially concerning the convergence properties of Q-learning within the LLM-based imagination space. Further proofs would add rigor to the framework's guarantees.

2. The mixed-examination mechanism, while useful, places a considerable dependency on LLM self-correction capabilities. Although current models like GPT-4 demonstrate adequate self-examination, reliance on this might limit robustness, especially if the model is deployed in new or noisy environments where the LLM might hallucinate.

3. The method has been evaluated on BlocksWorld and RoCo-benchmark, which are well-structured tasks. Further evaluations in more unstructured environments could provide insight into the adaptability of MLAQ in real-world scenarios with higher variability.

**Questions:**

1. Can you provide more theoretical justification or proofs for the convergence properties of Q-learning within the LLM-based imagination space? Specifically, under what conditions does MLAQ guarantee convergence to an optimal policy?

2. Given that MLAQ relies heavily on the LLM's self-examination for correcting imaginary transitions, how robust is the framework in handling LLM hallucinations? Have you explored scenarios where the LLM produces incorrect or inconsistent outputs, and if so, how does MLAQ mitigate this risk?

3. Could you discuss how sensitive MLAQ’s performance is to the hyperparameters used in the UCB variant (e.g., coefficients and confidence bounds)? Understanding this could help gauge the stability and generality of the method across different domains.

4. The MLAQ framework appears computationally intensive due to iterative Q-learning and the replay buffer management. Could you elaborate on the computational demands of MLAQ relative to traditional LLM agents? Are there strategies within the framework to reduce this overhead?

5. Have you tested MLAQ in more open-ended or unstructured environments, such as robotic tasks with real-world constraints (e.g., noisy sensors, dynamic obstacles)? If not, do you foresee any limitations or necessary modifications to make the model applicable to such environments?

6. Could you share insights on the types of tasks or domains where MLAQ may struggle to achieve optimal decision-making? Understanding these limitations can help contextualize the framework’s applicability.

7. Do you see potential for enhancing the LLM-based self-examination process, perhaps by combining it with non-LLM-based checks? For instance, could integrating environmental feedback more dynamically (e.g., through real-time adaptive learning) reduce reliance on the LLM’s introspective accuracy?

---

> ### Author Response · Authors · 2024-11-20
> **Authors Response (Part 1/2)**
>
> Dear Reviewer, thank you for your time to read our paper and provide useful comments. We carefully address your concerns as follows:
>
> ---
>
> **Q1: What are the scope and limitations of the MLAQ algorithm? Can MLAQ be applied in complex and unstructed environments?**
>
> **Scope.** MLAQ performs best in deterministic & fully observable scenarios. When applied to new scenarios, users only need to provide prompts for the LLM-based modules. Our experimental results have demonstrated this.
>
> **Experiments in complex environments.** MLAQ also performs well in complex environments with stochasticity and partial observability. We evaluate MLAQ on Crafter (a 2D version of Minecraft), an environment featuring high partial observability and complexity. The player has 16 sub-goals, and its final goal is to mine diamonds. Experimental results shows that MLAQ significantly outperforms existing algorithms. **Please refer to the overall response for more detail about this experiment.**
>
> **Limitations.** While the results on Crafter validates MLAQ's capability in handling complex, stochastic scenarios, we also identified a major limitation: MLAQ have limitations in addressing scenarios with stochasticity and partial observability.
> + For stochastic environments, the same state-action pair may lead to multiple next states, which significantly increases MLAQ's exploration demands in the imagination space, making it difficult for MLAQ to achieve optimal decision-making under limited test-time compute budget.
> + Although MLAQ outperforms other LLM agents, its performance remains lower than MLAQ-gt, which uses the ground truth dynamics function to eliminate stochasticity of Crafter. Please refer to the overall response for more details of the experiments.
>
> ---
>
> **Q2: To what extent can MLAQ's self-examination mechanism mitigate the hallucination problem in LLMs?**
>
> Hallucination of LLM is detrimental to MLAQ because the LLM-based world model may imagine wrong transitions. MLAQ mitigates its impact on optimal decision-making capability through three approaches, rather than solely relying on self-examination.
> + **Self-examination.** The first line of defense against erroneous outputs. It leverages the LLM to verify outputs. Results **demonstrate its nearly 99% accuracy in deterministic, fully observable domains (see Section 4.5)**. Nevertheless, erroneous transitions may still occur in memory, as evidenced by the non-zero (though very close to zero) Env Replans data in Table 2. In such cases, MLAQ employs other mechanisms.
> + **Env-examination.** For erroneous transitions emerged during imaginary interaction, environmental corrections can be made using ground-truth transitions to correct the memory and saved in the LLM-based module's prompt to prevent similar issues.
> + **RL-based optimization.** As mentioned in our manuscript, MLAQ's additional reliance on RL further enhances its robustness. Trajectories originating from error states mostly fail to complete the given task in Q-learning, resulting in significantly lower state values compared to correct states (unless multiple cascading errors coincidentally lead back to the correct trajectory, which has an extremely low probability).
>
> ---
>
> **Q3: What are the computational demands of MLAQ relative to traditional LLM agents?**
>
> Using terminology from LLM reasoning, MLAQ is a test-time compute approach in LLM-based decision-making, enhancing zero-shot optimal decision-making by combining the exploration in LLM-based imagination space with the RL-based optimization. The main computational demands include:
> + **Exploration in LLM-based imagination space (majority, about 95%).** MLAQ's planning in the imagination space is computationally efficient due to its UCB variant and transition re-utilization.
>   + **Re-utilization of imagination results.** 1) MLAQ re-utilizes transitions in memory during planning to **avoid redundant imagination**. 2) MLAQ interacts with the environment by using Q-learning-derived argmax actions, **eliminating the need to query the LLM** per step.
>   + **Efficiency.** MLAQ is an efficient trade-off between basic policy and MCTS. It avoids unnecessary exploration when the base policy is already optimal, matching transitional agents' costs. For weaker policies, it uses UCB-guided exploration to improve performance. Compared to MCTS (in RAP), MLAQ is more similar to Dreamer, which uses Q-learning for computing state values.
>   + Compared to traditional LLM agents, MLAQ requires more exploration than search-free methods like RoCo for optimal decisions. However, results in Table 2 shows that agents like RoCo requires higher token usage due to their low correctness in decision-making.
> + **Q-learning Process (< 5%).** Given that natural language tasks typically have small state spaces (rarely exceeding 10,000 states), MLAQ employs a tabular approach for storing transitions and executing Q-learning iterations. This computational demand is negligible compared to the planning phase.

---

> ### Author Response · Authors · 2024-11-20
> **Authors Response (Part 2/2)**
>
> **Q4: Can authors provide more theoretical analysis about MLAQ's convergence?**
>
> MLAQ employs the Q-learning algorithm, whose convergence has been well established. We conjecture that the reviewer is interested in the **performance gap between the converged policy in the true environment versus that in the imagination space**, which depends on the accuracy of the LLM-based world model.
>
> Analyzing MLAQ's convergence at the token level is quite complex. Given that MLAQ is a **Dreamer-style model-based RL method**, we can directly apply theories from previous works. Drawing from Theorem 4.1 in [1], we present the following conclusion.
>
> **Bound Theorem.** Let the expected TV-distance between two transition distributions be bounded at each timestep by $\epsilon_m$ and the policy divergence be bounded by $\epsilon_\pi$. Then **the true returns and model returns of the policy are bounded as**:
>
> $\eta[\pi] \ge \hat{\eta}[\pi] - [\frac{2\gamma(\epsilon_m+2\epsilon_\pi)}{(1-\gamma)^2}+\frac{4r_{\rm max}\epsilon_\pi}{(1-\gamma)}]$
>
> where $\eta$ is the returns of the policy in the true environment, $\hat{\eta}$ is the returns of the policy under the LLM-based world model. $r_{\rm max}$ is the maximum value of rewards, $\epsilon_m = \max_{(s, a) \sim \mathcal{M}} [D_{\rm TV}(\mathbb{T}(\cdot|s, a))||(\mathbb{T}(\cdot|s, a; \theta))]$ is the **generalization error** of the LLM-based world model, $(s, a)$ is sampled from **MLAQ's memory module** $\mathcal{M}$, and $\mathbb{T}(\cdot|s, a))$ and $\mathbb{T}(\cdot|s, a; \theta))$ are the **transition distribution** of the true environment and LLM-based world model, respectively. $\epsilon_\pi \ge D_{\rm TV}(\pi||\pi_\mathcal{M})$ is the policy divergence between greedy policy $\pi$ derived from Q-learning (output argmax actions) and the data-collecting policy from the memory $\pi_\mathcal{M}$.
>
> From the definitions of $\epsilon_\pi$ and $\epsilon_m$, we can observe that **$\epsilon_\pi$ increases as $\pi$ deviates further from $\pi_\mathcal{M}$, while $\epsilon_m$ increases as the domain becomes more challenging to simulate (larger prediction errors in LLM-based world model)**. Consequently, the greedy policy derived from Q-learning yields a larger return gap when interacting with the true MDP (environment), indicating that policies optimal in the imagination space may not be optimal in the true MDP. MLAQ primarily mitigates the bias between imagination space and true MDP by preventing excessive $\epsilon_m$ values through mixed-examination.
>
> [1] Janner, Michael, et al. "When to trust your model: Model-based policy optimization." NeurIPS, 2019.
>
> ---
>
> **Q5: Can self-examination use a non-LLM-based Checker?**
>
> Of course! Actually, both action and prediction verification can be implemented using non-LLM-based checkers. However, training them would **require substantial ground-truth data**, which contradicts MLAQ's zero/few-shot motivation. If users **do not specifically require this**, non-LLM-based checkers might be more effective.
>
> ---
>
> **Q6: How sensitive is MLAQ's performance to the hyperparameter used in the UCB variant (e.g., coefficients and confidence bounds)?**
>
> I would like to clarify to the reviewers that **the only hyperparameter in our UCB variant** is $w$, while the form of confidence bound remains consistent with the original UCB. Furthermore, we tested scenarios with $w$ values of **1, 4 (orginal value), and 16** in the 2-step BlocksWorld environment, with results as follows.
>
> | w | 1 | 4 | 16 |
> | - | - | - | - |
> | Optimal rate | 1.00	| 1.00 | 1.00 |
> | Token | 41123 | 36238 | 60328 |
>
> The results indicate that different values of $w$ **have small impact on the optimal rate**. However, significant differences are observed in **token consumption**. When $w$ equals 1, the token consumption is comparable to when $w$ equals 4. However, when $w$ increases to 16, the token consumption is approximately 1.5 times higher than cases where $w$ is 1 or 4. This is primarily attributed to the fact that **with larger w values, the planning process tends to favor exploration of new actions** rather than exploiting existing actions.
>
> Furthermore, table 9 in the appendix presents other hyperparameters of MLAQ. In table 9, only the environmental horizon varies across different domains, while other hyperparameters do not require tuning. However, users can adjust these parameters according to their computational budget and problem complexity.

---

> ### Author Response · Authors · 2024-11-24
> **Kindly Inquire Whether We Have Addressed Your Concerns**
>
> Dear Reviewer,
>
> **Thank you** once again for taking the time to review our manuscript and for considering our responses to your comments.
>
> If you require any further clarification or would like additional points to be included in our response, we welcome any further discussion to ensure everything is clear and satisfactory. If we have adequately addressed your concerns, we would sincerely appreciate your **re-evaluation** and **reconsideration** of your rating.
>
> Thank you for your consideration.
>
> Best regards,
>
> Authors

---

### Author Response · Authors · 2024-11-20
**Overall response for all reviewers**

Firstly, we extend our gratitude to **ALL** reviewers for their time and effort in reviewing this manuscript. We are particularly grateful for their positive feedback, as three reviewers acknowledged that the paper is **well-written and provides a comprehensive description of the methodology**, and most reviewers acknowledged the valuable innovations presented in this work and recognized our thorough experimental validation of **MLAQ's zero- and few-shot optimal decision-making capabilities**. We have addressed the reviewers' concerns by supplementing more descriptions as suggested, which we hope will help better understand MLAQ's contributions. We look forward to a positive re-evaluation based on these revisions.

During the rebuttal period, we provide **two new information** about MLAQ for the reviewers:
- Drawing from MBPO's theoretical framework, we provide a **theoretical analysis of the converged policy** when LLM-based world models contain errors
- We evaluate MLAQ on the **much more complex and open-ended Crafter environment**, comparing it with existing LLM agents and RL agents. Results indicate that while MLAQ can be effectively applied to stochastic, partially observable environments, it is more suitable for deterministic, fully observable settings (**see "Experiments on Crafter" for more detail**).

Additionally, both the GitHub repository (https://github.com/laq2024/MLAQ/) and interactive demonstration website (http://mlaq.site) are completely anonymized, created through newly registered accounts specifically for this conference. We affirm that no personal information has been included in either link.

---

### Author Response · Authors · 2024-11-20
**Experiments on Crafter**

Many reviewers mentioned more experimental results of MLAQ in complex scenarios. We first categorize scenarios into two types to evaluate their complexity:
+ **Deterministic & fully observable**. RoCo and BlocksWorld belong to this category, where the state information for LLM is fully-known, and executing an action can only lead to one deterministic next state (when moving a cube from A to B, the cube can only appear at B rather than at another random place C). This type of scenario can cover most practical applications.
+ **Stochastic & partially observable (Much more complex)**. This category involves partially observable settings in open-ended environments, such as in Minecraft, where the overall map information is unknown, and player exploration on the map is partially observable (for example, when digging downward, players don't know whether the next cube is stone or coal).

To validate MLAQ's performance in partially observable scenarios with high complexity, we chose **Crafter[1] (a 2D version of MineCraft)** as our experimental environment.
+ In Crafter, players need to mine diamonds on a 64*64 map, and there are 16 sub-goals to complete (collecting woods, stones, make wood pickaxe, etc.), making it **a challenging task with extremely long decision sequences**. Additionally, the reward signals are only provided when successfully achieving a sub-goal.
+ Players can only observe a 9 * 9 local map, and we adopt SmartPlay's natural language observation setting: for multiple items in the view, only the nearest one will be shown in the **observation**. For example, even if there are 3 trees in the view, the observation will only show "tree is 3 steps to your north-east", which further intensifies the partial observability of Crafter.

Since the same state-action pair may lead to different next states for **stochastic domains**, we made two minor modifications to MLAQ:
+ We remove the Prediction Checker, as it's difficult to provide effective verification in the cases with stochasticity.
+ We remove MLAQ's re-utilization of memory transitions during planning, because direct re-utilization would lose stochasticity for **potential multiple next states**. However, this can be adjusted as needed in practical, such as starting random re-utilization when the number of next states exceeds a threshold.

**Here are the experimental results**:

| Methods | Human[1] | MLAQ-gt | MLAQ-script | MLAQ | DiVE[3] | EnvGen[4] | Dreamer V3[5] | PPO[6] | Random [1] |
| - | - | - | - | - | - | - | - | - | - |
| Scores | 50.5 | 46.2 | 42.6 | 39.9 | 35.9 | 32.2 | 14.5 | 4.6 | 1.6 |

We calculate the scores using the same methodology as existing works, which is determined by the ratio of unblocked achievements. The data of MLAQ are obtained using the same testing methods as SmartPlay[2], while other data are sourced from their original papers or from [1]. In Crafter, **the move action exhibits strong stochasticity**, making it impossible for players to infer the next observation.

Therefore, we conduct three experiments where the only difference lies in how the world model predict next observations **for move actions**, while all other modules **remain consistent** with the original MLAQ.
+ **MLAQ-gt** utilizes ground-truth environment simulation for move actions, providing the exact next observation (next state) for given observation-action (state-action) pair, thus effectively **eliminating movement stochasticity**.
+ **MLAQ-script** generates next observations for move actions through **scripted randomization** (e.g., after moving north, nine unknown areas appear at the northernmost position, with randomly assigned items like stone, sand).
+ **MLAQ** reconstructs the aforementioned script into prompts, allowing LLM to fully implement an **LLM-based world model**.

The results align with expectations: MLAQ-gt achieves best performance using ground-truth environment for move actions, consistently crafting iron pickaxes but struggling to explore the location of diamonds within limited steps in 64 * 64 maps. MLAQ-script and MLAQ **outperform other existing LLM agents and RL agents**, but significantly below MLAQ-gt, indicating stochasticity substantially impacts MLAQ. These experimental data are averaged from eight tests, and we commit to supplementing more comprehensive data in the camera-ready version and incorporating it into the revised manuscript.

[1] Hafner, Danijar. "Benchmarking the Spectrum of Agent Capabilities." ICLR, 2022.

[2] Wu, Yue, et al. "SmartPlay: A Benchmark for LLMs as Intelligent Agents." ICLR, 2024.

[3] Sun, Zhiyuan, et al. "Enhancing Agent Learning through World Dynamics Modeling." EMNLP Findings, 2024.

[4] Zala Abhay, et al. "EnvGen: Generating and Adapting Environments via LLMs for Training Embodied Agents." COLM, 2024.

[5] Hafner, Danijar, et al. "Mastering diverse domains through world models." arXiv, 2023.

[6] Schulman, John, et al. "Proximal policy optimization algorithms." arXiv, 2017.

---

### Meta-Review · Area_Chair_oqCk · 2024-12-20

**Metareview:**

The core contribution of this work lies in leveraging LLMs as a world model to generate imaginary transitions for Q-learning. MLAQ includes a UCB-inspired planner and a mixed-examination mechanism to balance exploration, enhance transition quality, and achieve sub-linear regret bounds. Experiments demonstrate MLAQ's efficacy in benchmarks like BlocksWorld and Crafter.

The idea of using an LLM as a world model/simulator is forward-looking. Despite being an obvious idea, little prior work has shown effectiveness in this approach, so the authors' contribution is valuable. In a long run, LLM-based simulation eliminates the need for extensive environmental interactions. Empirical results show strong performance, particularly on long-horizon decision-making benchmarks.

The biggest criticism lies in writing clarity. Some reviewers found the paper difficult to follow, citing unclear problem statements, undefined terms, and missing implementation details. "Empowering LLM agents with zero- or few-shot optimal decision-making capabilities" is too generic of a problem statement. Concerns were also raised about the validity of the zero-shot claim due to the mixed-examination mechanism requiring some environmental feedback.

The authors are expected to revise the paper and improve the writing clarity significantly in the final submission according to reviewers feedback. It will be interesting to see the effectiveness of the proposed approach in more complex and practical benchmarks.

**Additional Comments On Reviewer Discussion:**

During the rebuttal period, reviewers raised concerns about the paper's clarity, the validity of its zero-shot claims, and the limited experimental scope. Reviewers XLz6 and ZK4E criticized the writing quality and the lack of detailed explanations for core concepts such as the imagination space and mixed-examination mechanism, while also questioning the reliance on environmental feedback in a purportedly zero-shot framework. The authors responded with additional experiments in Crafter, clarifications on terminology, and an extended discussion of MLAQ's generalizability and trade-offs between zero- and few-shot operation. They also explained the differences between MLAQ and related works like Decision Transformer, addressing concerns about the paper's novelty. While reviewers ouUC, 4yYK, and FSC6 were satisfied with these clarifications and recognized the paper's contributions, XLz6 and ZK4E maintained their reservations about writing quality and implementation clarity. Balancing these perspectives, I think that the paper's contributions and empirical results outweighed the remaining concerns, leading to an acceptance recommendation with suggested revisions for clarity and scope.

---

> ### Public Comment · ~Jiajun_Chai1 · 2025-02-28
> **Notes on the Camera-Ready Version**
>
> Dear Area Chair and Readers,
>
> In the Camera-Ready version, we have revised the original text according to the suggestions from the reviewers and the Area Chair. The main modifications include clarifying unclear expressions and providing necessary definitions for mentioned terms or variables, hoping that readers can fully comprehend the contributions of this paper. Additionally, experiments conducted on Crafter have been added to Appendix E, which effectively demonstrate the effectiveness of the proposed approach in more complex and practical benchmarks.
>
> Lastly, we would like to express our gratitude to all the reviewers and the Area Chair for dedicating their time and effort to our paper. We also hope that all readers in the community will engage with us in discussions, aiming to further enhance the optimal decision-making capabilities of LLM agents in future work.
>
> Sincerely,
>
> Authors

---

### Decision · Program_Chairs · 2025-01-22

Accept (Poster)